# Multi-Agent First Order Constrained Optimization in Policy Space

**Youpeng Zhao**[1], **Yaodong Yang**[2][†], **Zhenbo Lu**[3][†], **Wengang Zhou**[1,3], **Houqiang Li**[1,3]

[1]University of Science and Technology of China, [2]Institute for AI, Peking University
[3]Institute of Artificial Intelligence, Hefei Comprehensive National Science Center
`zyp123@mail.ustc.edu.cn, yaodong.yang@pku.edu.cn`
`luzhenbo@iai.ustc.edu.cn, {zhwg,lihq}@ustc.edu.cn`

## Abstract

In the realm of multi-agent reinforcement learning (MARL), achieving high performance is crucial for a successful multi-agent system. Meanwhile, the ability to avoid unsafe actions is becoming an urgent and imperative problem to solve for real-life applications. Whereas, it is still challenging to develop a safety-aware method for multi-agent systems in MARL. In this work, we introduce a novel approach called Multi-Agent First Order Constrained Optimization in Policy Space (MAFOCOPS), which effectively addresses the dual objectives of attaining satisfactory performance and enforcing safety constraints. Using data generated from the current policy, MAFOCOPS first finds the optimal update policy by solving a constrained optimization problem in the nonparameterized policy space. Then, the update policy is projected back into the parametric policy space to achieve a feasible policy. Notably, our method is first-order in nature, ensuring the ease of implementation, and exhibits an approximate upper bound on the worst-case constraint violation. Empirical results show that our approach achieves remarkable performance while satisfying safe constraints on several safe MARL benchmarks.

## 1 Introduction

Cooperative multi-agent systems play a significant role in various domains, where a group of agents coordinate with each other to accomplish tasks and collaboratively optimize cumulative rewards for the team[1, 2]. Such a setting is frequently employed in many real-life scenarios such as robotics [3], autonomous vehicles [4], traffic light control [5] and the smart grid [6].Thanks to the recent remarkable advance of reinforcement learning techniques in various complex tasks [7–9], multi-agent reinforcement learning (MARL) has attracted substantial attention and quite a few algorithms have been proposed, including value-based methods [10–14] and policy gradient methods [15–18]. Despite the notable achievement in academia, most MARL algorithms prioritize policy optimization solely for reward maximization, while disregarding potential negative or harmful consequences resulting from the agents' behaviors. Consequently, these methods can not be directly deployed in practice. In reality, many applications often require the agents to refrain from taking certain actions or visiting particular states [19, 20]. For instance, an unmanned car must adhere to traffic regulations by not crossing a red light, even while pursuing its destination, in order to prioritize safety.

To address the above issues, researchers have been devoted to developing algorithms that learn policies which adhere to safety constraints and great progress has been made in single-agent RL setting[20–23]. However, it is still a daunting challenge to develop safe policies for multi-agent systems. Because of the existence of multiple agents, the environment may suffer from non-stationarity due to simultaneously learning agents, posing a non-negligible challenge to the training process. Moreover, ensuring safety in MARL is highly intricate. To provide a better depiction of the inherent challenges,

---

[†]Corresponding authors: Yaodong Yang and Zhenbo Lu

37th Conference on Neural Information Processing Systems (NeurIPS 2023).

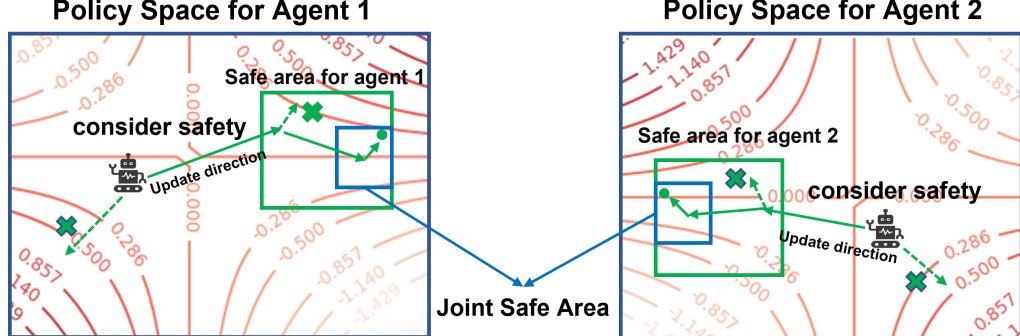

Figure 1: An example to illustrate the challenges in safe MARL. Reward distribution is presented in the policy space for each agent. Green rectangles delineate the individual safe areas for each agent and the blue ones depict the joint safe area. Green arrows symbolize the update direction for each agent whereas the dashed ones signify the intended update direction if each agent solely considers its own objectives and the crosses represent that the dashed update direction is unsuitable in safe MARL.

we show an illustrative example in Figure 1. This figure showcases the policy space for each agent, wherein the reward distribution is presented. The green rectangles delineate the individual safe areas for each agent and the blue ones depict the joint safe area. The green arrows symbolize the update direction for each agent, whereas the dashed ones signify the intended update direction if each agent solely considers its own objectives. When agents are randomly initialized, they naturally tend to update the policy along the dashed line arrows to maximize the reward. However, due to the presence of safety constraints, they must adjust their updates along the solid line arrows to ensure their policies fall within the safe area. This is non-trivial as it may conflict with their goal of reward maximization. Furthermore, after individually addressing their safety constraints, the agents must also consider the safety constraints of others to collectively converge to the joint safe area, adding more complexity to the optimization process for the entire system. The interplay of these factors underscores the inherent difficulties in solving the problem of safe MARL.

Recently, as a notable advancement in the field of safe MARL, Multi-Agent Constrained Policy Optimization (MACPO) [24] has been proposed as a safe and effective solution. MACPO attains the properties of both monotonic improvement guarantee and safety constraints satisfaction guarantee at every iteration during training. However, this algorithm involves solving an optimisation problem using Taylor approximations and inverting a high-dimensional Fisher information matrix. As a result, the computation is very complex and the policy update often becomes infeasible. To alleviate this issue, it requires additional recovery steps, which, however, sometimes causes updates to be backtracked and samples to be wasted.

Taking inspiration from the sequential policy update scheme introduced in HATRPO [18], we have devised a simpler approach to incorporate safety constraints in solving safe MARL problems compared with MACPO. The resulting algorithm, Multi-Agent First Order Constrained Optimization in Policy Space (MAFOCOPS), aims to address the following question of how to achieve the best constraint-satisfying policy update given the current policy for each agent. Our method follows a two-step process to give a solution to this problem. First, based on the theoretical foundations presented in MACPO, we demonstrate that the best policy update has a near-closed form solution when attempting to solve the optimal policy in the nonparametric policy space. Subsequently, we project the policy back into the parametric policy space as direct evaluation of the optimal policy is usually not feasible. This can be realized by sampling from the current policy and evaluating a loss function between the parameterized policy and the optimal policy obtained in the nonparametric policy space. Notably, our algorithm only employs first-order approximations, making it straightforward to implement, and has an approximate upper bound for worst-case constraint violation. To validate the effectiveness of our approach, we conduct experiments on two safe MARL benchmarks proposed by [24], namely Safe MAMuJoCo and Safe MAIG. The experimental results demonstrate the superior performance of MAFOCOPS compared to MACPO, despite being a simpler algorithm.

## 2 Related Work

Safety has become a crucial and longstanding concern in the field of reinforcement learning [25]. In this section, we discuss recent advancements in the domain of safe RL for multi-agent environments.

The realm of safe multi-agent reinforcement learning is a nascent area of research that has gained increasing importance [26]. Several attempts have been made to tackle safe MARL, but most existing approaches have limitations. For example, CMIX [27] leverages the value function decomposition method and modifies the reward function to account for constraint violations, yet this algorithm fails to provide safety guarantees during training. Another approach is Safe Dec-PG [28] which proposes a decentralized policy gradient descent-ascent method by means of a consensus network and employs a primal-dual framework to balance reward maximization and cost minimization. Nevertheless, the consensus requirement in this work equivalently imposes an extra constraint of parameter sharing among neighbouring agents, potentially leading to sub-optimal policy [18]. It is noteworthy that most current safe MARL methods are tailored to robotics tasks, utilizing techniques such as barrier certificates [29, 30] or model predictive shielding [31] to address safety issues. These methods, however, are specifically designed for robotics applications. Besides, they often require supervised learning based approaches or specific assumptions concerning the state space and environment dynamics.

As a recent remarkable solution, MACPO [24] incorporates a sequential policy update scheme. This algorithm develops the multi-agent trust region learning based on CPO [20] and provides theoretical guarantees of both monotonic improvement in reward and compliance with cost constraints in a multi-agent setting. To achieve practical solutions to the safe MARL problems, the policy for each agent needs to be parameterized with a neural network, which effectively represents the policy in policy space. However, achieving parameterized policies in MACPO involves solving optimization problems using first and second-order Taylor approximation, which includes taking the inverse of Fisher information matrix, and the computation is implemented by the conjugate gradient method [32]. These operations can introduce nonnegligible approximation errors, thereby compelling MACPO to undertake additional steps during each update in the training process in order to recover from constraint violations. In contrast, our algorithm takes a different approach by solving the optimization problem within the nonparametric space and then projecting the results back into the parameter space. By leveraging a simple first-order method to eliminate the approximation error, our algorithm ultimately outperforms MACPO.

## 3 Problem Formulation

A safe MARL problem can be formulated as a Constrained Markov Decision process, which is described by a tuple denoted as $< \mathcal{N}, \mathcal{S}, \mathcal{A}, p, \rho^0, \gamma, R, \boldsymbol{C}, c >$. Here, $\mathcal{N} = \{1, 2, \cdots, n\}$ denotes the set of agents involved in the system, $\mathcal{S}$ is the state space, $\mathcal{A} = \Pi_{i=1}^n \mathcal{A}^i$ represents the product of the action spaces of agents, *i.e.,* joint action space, $p : \mathcal{S} \times \mathcal{A} \times \mathcal{S} \to \mathbb{R}$ is the probabilistic transition function, $\rho^0$ is the initial state distribution and $\gamma \in [0, 1)$ is the discounted factor. The team reward function $R : \mathcal{S} \times \mathcal{A} \to \mathbb{R}$ maps state and joint actions to a scalar reward while $\boldsymbol{C} = \{C_j^i\}_{1 \leq j \leq m^i}^{i \in \mathcal{N}}$ is the set of sets of cost functions denoted in the form $C_j^i : \mathcal{S} \times \mathcal{A}^i \to \mathbb{R}$ (every agent $i$ has $m^i$ cost functions). The set of corresponding cost-constraining values is given by $c = \{c_j^i\}_{1 \leq j \leq m^i}^{i \in \mathcal{N}}$. At time step $t$, the multi-agent system is situated in state $s_t$ and every agent $i$ selects an action $a_t^i$ based on its policy $\pi^i(a^i|s)$, forming a joint action $\boldsymbol{a_t} = (a_t^1, \cdots, a_t^n)$ and joint policy $\boldsymbol{\pi}(\boldsymbol{a}|s) = \Pi_{i=1}^n \pi^i(a^i|s)$. This leads the environment to transit to a new state $s_{t+1} \sim p(\cdot|s_t, \boldsymbol{a_t})$ according to the probabilistic transition function and the system receives the reward $R(s_t, a_t)$ while each agent $i$ incurs individual costs $C_j^i, \forall j = 1, \cdots, m^i$. This study focuses on a fully-cooperative multi-agent setting where all agents share the same reward function, aimed at maximizing the expected total reward:

$$J(\boldsymbol{\pi}) \triangleq E_{s_0 \sim \rho^0, \boldsymbol{a}_{0:\infty} \sim \boldsymbol{\pi}, s_{1:\infty} \sim p}[\sum_{t=0}^{\infty} \gamma^t R(s_t, \boldsymbol{a}_t)].$$

Moreover, we impose that each agent $i$ satisfies its safety constraint, which is defined as

$$J_j^i(\boldsymbol{\pi}) \triangleq E_{s_0 \sim \rho^0, \boldsymbol{a}_{0:\infty} \sim \boldsymbol{\pi}, s_{1:\infty} \sim p}[\sum_{t=0}^{\infty} \gamma^t C_j^i(s_t, a_t^i)] \leq c_j^i, \forall j = 1, \cdots, m^i.$$

We define the state-action value and the state-value function in terms of reward as

$$Q_{\boldsymbol{\pi}}(s, \boldsymbol{a}) \triangleq E_{s_{1:\infty} \sim p, \boldsymbol{a}_{1:\infty} \sim \boldsymbol{\pi}}[\sum_{t=0}^{\infty} \gamma^t R(s_t, \boldsymbol{a}_t)|s_0 = s, \boldsymbol{a}_0 = \boldsymbol{a}], V_{\boldsymbol{\pi}}(s) \triangleq E_{\boldsymbol{a} \sim \boldsymbol{\pi}}[Q_{\boldsymbol{\pi}}(s, \boldsymbol{a})].$$

It's worth noting even though the action $a_t^i$ taken by agent $i$ does not directly impact the costs $\{C_j^k(s_t, a_t^k)\}_{j=1}^{m_k}$ of other agents $k \neq i$ from the above formulation, the total costs can still be influenced by this action implicitly due to its influence on the subsequent state $s_{t+1}$. This formulation captures the realistic multi-agent interactions in real world. For instance, when a car runs a red light, although other cars may not be immediately endangered by this action, the resulting disruption in traffic flow may lead to potential hazards later on. To illustrate the $j_{th}$ cost function of agent $i$, we express the corresponding state-action cost value function and the state cost value function as below:

$$Q_{j,\boldsymbol{\pi}}^i(s, \boldsymbol{a}) \triangleq E_{\boldsymbol{a}^{-i} \sim \boldsymbol{\pi}^{-i}, s_{1:\infty} \sim p, \boldsymbol{a}_{1:\infty} \sim \boldsymbol{\pi}}[\sum_{t=0}^{\infty} \gamma^t C_j^i(s_t, a_t^i)|s_0 = s, a_0^i = a^i],$$

$$V_{j,\boldsymbol{\pi}}^i(s) \triangleq E_{\boldsymbol{a} \sim \boldsymbol{\pi}, a \sim \boldsymbol{\pi}}[\sum_{t=0}^{\infty} \gamma^t C_j^i(s_t, a_t^i)|s_0 = s].$$

Notably, although similar in form to traditional $Q_{\boldsymbol{\pi}}$ and $V_{\boldsymbol{\pi}}$, the cost value function $Q_{j,\boldsymbol{\pi}}$ and $V_{j,\boldsymbol{\pi}}$ involve additional indices $i$ and $j$, where the subscript $i$ refers to an agent and $j$ denotes the $j^{th}$ cost.

Motivated by the sequential policy update scheme, we pay close attention to determining the contribution of different subsets of agents to overall performance in this study. We denote an arbitrary subset $\{i_1, \cdots, i_h\}$ of agents as $i_{1:h}$ while $-i_{1:h}$ refers to its complement. Given the agent subset $i_{1:h}$, we define the multi-agent state-action value function:

$$Q_{\boldsymbol{\pi}}^{i_{1:h}}(s, \boldsymbol{a}^{i_{1:h}}) \triangleq E_{\boldsymbol{a}^{-i_{1:h}} \sim \boldsymbol{\pi}^{-i_{1:h}}}[Q_{\boldsymbol{\pi}}(s, \boldsymbol{a}^{i_{1:h}}, \boldsymbol{a}^{-i_{1:h}})].$$

Furthermore, for disjoint sets $i_{1:h}$ and $j_{1:k}$, the multi-agent advantage function is defined as follows:

$$A_{\boldsymbol{\pi}}^{i_{1:h}}(s, \boldsymbol{a}^{j_{1:k}}, \boldsymbol{a}^{i_{1:h}}) \triangleq Q_{\boldsymbol{\pi}}^{j_{1:k}, i_{1:h}}(s, \boldsymbol{a}^{j_{1:k}}, \boldsymbol{a}^{i_{1:h}}) - Q_{\boldsymbol{\pi}}^{j_{1:k}}(s, \boldsymbol{a}^{j_{1:k}}).$$

An interesting and critical observation concerning the aforementioned multi-agent advantage function is that the advantage $A_{\boldsymbol{\pi}}^{i_{1:h}}$ can be written as the sum of sequentially-unfold multi-agent advantages of individual agents, that is,

**Lemma 1** (Multi-agent advantage decomposition [18]) For any state $s \in \mathcal{S}$, subsets of agents $i_{1:h} \in \mathcal{N}$ and joint action $\boldsymbol{a}^{i_{1:h}}$, the following identity holds

$$A_{\boldsymbol{\pi}}^{i_{1:h}}(s, \boldsymbol{a}^{i_{1:h}}) \triangleq \sum_{j=1}^{h} A_{\boldsymbol{\pi}}^j(s, \boldsymbol{a}^{i_{1:j-1}}, a^{i_j}).$$

## 4 Method

Expanding on the above foundational concepts and the derivatives of multi-agent trust region learning with constraints, MACPO provided an important insight. It highlighted that when the policy changes for all agents are sufficiently small, each agent can learn a better policy $\bar{\pi}^i$ by only considering its own surrogate return and costs, which is consistent with the sequential policy update scheme. In our work, building upon the formulas in MACPO, we can deduce that, for agent $i_h$ and the index of its cost functions $j$, given the joint policy $\boldsymbol{\pi}_{\boldsymbol{\theta}_k}$ at the $k$th iteration and updated policies of the previous agent sets $i_{1:h-1}$, namely $\pi_{\theta_{k+1}}^{i_{1:h-1}}$, the new policy is obtained by solving the following optimization problem:

$$\underset{\pi_\theta^{i_h}}{maximize} \, E_{s \sim \rho_{\boldsymbol{\pi}_{\boldsymbol{\theta}_k}}, a^{i_{1:h-1}} \sim \pi_{\theta_{k+1}}^{i_{1:h-1}}, a^{i_h} \sim \pi^{i_h}}[A_{\boldsymbol{\pi}_{\boldsymbol{\theta}_k}}^{i_h}(s, a^{i_{1:h-1}}, a^{i_h})], \quad (1)$$

$$s.t. J_j^{i_h}(\boldsymbol{\pi}_{\boldsymbol{\theta}_k}) + E_{s \sim \rho_{\boldsymbol{\pi}_{\boldsymbol{\theta}_k}}, a^{i_h} \sim \pi^{i_h}}[A_{j, \boldsymbol{\pi}_{\boldsymbol{\theta}_k}}^{i_h}(s, a^{i_h})] \leq c_j^{i_h}, \forall j \in 1, \cdots, m^{i_h}, \quad (2)$$

$$\bar{D}_{KL}(\pi_\theta^{i_h}, \pi_{\theta_k}^{i_h}) \leq \delta. \quad (3)$$

where $\bar{D}_{KL}(\pi_\theta^{i_h}, \pi_{\theta_k}^{i_h}) \triangleq E_{s \sim \rho_{\boldsymbol{\pi}_{\boldsymbol{\theta}_k}}}[D_{KL}(\pi_{\theta_k}^{i_h}(\cdot|s), \pi_\theta^{i_h}(\cdot|s))]$. A simple proof is presented in Appendix A. When solving the optimization problem (1-3), we use a two-step approach:

1. For agent $i_h$, when provided with the joint policy $\pi_{\theta_k}^-$ and updated policy $\pi_{\theta_{k+1}}^{i_{1:h-1}}$, find an *optimal update policy* $\pi^{i_h*}$ in the nonparameterized policy space, denoted by $\Pi$.

2. Project the policy found in previous step back into parameterized policy space $\Pi_\theta$ by solving for the closest policy $\pi_\theta \in \Pi_\theta$ to obtain $\pi_{\theta_{k+1}}^{i_h}$. Here we consider the set of parameterized policies $\Pi_\theta = \{\pi_\theta : \theta \in \Theta\} \in \Pi$ which allows for evaluation and sampling.

## 4.1 Finding the Optimal Update Policy

In the first step, we aim to find the optimal nonparameterized policy and propose the following solutions (see Appendix B for proof):

**Theorem 1.** For agent $i_h$, we define $b_j^{i_h} = c_j^{i_h} - J_j^{i_h}(\pi_{\theta_k})$. If $\pi_{\theta_k}$ is a feasible policy, the optimal policy takes the form

$$\pi^{i_h*}(a|s) = \frac{\pi_{\theta_k}^{i_h}(a|s)}{Z_{\lambda_j,\nu_j}(s)} exp\{\frac{1}{\lambda_j}(\eta_{\pi_{\theta_k}}(s,a^{i_h}) - \nu_j A_{j,\pi_{\theta_k}}^{i_h}(s,a^{i_h}))\}, \tag{4}$$

where $\eta_{\pi_{\theta_k}}(s,a^{i_h}) = E_{a^{i_{1:h-1}} \sim \pi_{\theta_{k+1}}^{i_{1:h-1}}}[A_{\pi_{\theta_k}}^{i_h}(s,a^{i_{1:h-1}},a^{i_h})], Z_{\lambda_j,\nu_j}(s)$ is the partition function that ensures Equation 4 to be a valid probability distribution, and $\lambda_j$ as well as $\nu_j$ are solutions to the following optimization problem:

$$\min_{\lambda_j,\nu_j \geq 0} \lambda_j \delta + \nu_j b_j^{i_h} + \lambda_j E_{s \sim \rho_{\pi_{\theta_k}}, a^{i_h} \sim \pi^{i_h*}}[log Z_{\lambda_j,\nu_j}(s)]. \tag{5}$$

The structure of the optimal policy exhibits an intuitive nature, as it assigns a substantial probability mass to regions of the state-action space with high returns. This allocation is counterbalanced by a penalty term multiplied by the cost advantage. Another desirable property of the optimal update policy is that for feasible policy $\pi_{\theta_k}$, it has an upper bound for worst-case guarantee for cost constraint satisfaction according to Lemma.2 in MACPO [24].

## 4.2 Approximating the Optimal Update Policy

When addressing the optimization problem 1-3 in the first step, the optimal update policy for agent $i_h$ may not necessarily reside within parameterized policy space $\Pi_\theta$. Consequently, evaluating or sampling from this policy becomes unfeasible. To this end, in the second step, we need to project the optimal update policy back into the parameterized policy space by minimizing the loss function:

$$L(\theta) = E_{s \sim \rho_{\pi_{\theta_k}}}[D_{KL}(\pi_\theta^{i_h} || \pi^{i_h*})(s)]. \tag{6}$$

In this context, $\pi_\theta^{i_h} \in \Pi_\theta$ represents some projected policy that serves as an approximation of the optimal update policy. To minimize this loss function, first-order methods can be employed. In doing so, we can leverage the following result as a useful tool in our optimization efforts:

**Corollary 1.** The gradient of $L(\theta)$ takes the form

$$\nabla_\theta L(\theta) = E_{s \sim \rho_{\pi_{\theta_k}}}[\nabla_\theta D_{KL}(\pi_\theta^{i_h} || \pi^{i_h*})[s]], \tag{7}$$

where

$$\nabla_\theta D_{KL}(\pi_\theta^{i_h} || \pi^{i_h*})[s] = \nabla_\theta D_{KL}(\pi_\theta^{i_h} || \pi_{\theta_k}^{i_h}) - \frac{1}{\lambda_j} E_{a \sim \pi_{\theta_k}^{i_h}}[\frac{\nabla_\theta \pi_\theta^{i_h}(a|s)}{\pi_{\theta_k}^{i_h}(a|s)}(\eta_{\pi_{\theta_k}}(s,a^{i_h}) - \nu_j A_{j,\pi_{\theta_k}}^{i_h}(s,a^{i_h}))]. \tag{8}$$

The proof is shown in Appendix C.

It's to be noted that 27 can be estimated by sampling from the trajectories generated by policy $\pi_{\theta_k}$, which allows us to train our policy using stochastic gradients, a key aspect of our methodology. This corollary serves as a guiding framework for our algorithm. At every iteration, we begin with a policy $\pi_{\theta_k}$, utilizing it to generate trajectories and collect relevant data. Subsequently, we employ this data in conjunction with 5 to estimate $\lambda_j$ and $\nu_j$. We then draw a minibatch from the collected data to estimate $\nabla_\theta L(\theta)$ as outlined in Corollary 1. After taking a gradient step using Equation 27, we draw another minibatch and repeat the process.

## 4.3 Practical Implementation

For agent $i_h$, solving the dual problems presented in 5 is computationally challenging when dealing with large state/action spaces as calculating the partition function $Z_{\lambda_j, \nu_j}(s)$ often involves evaluating a high-dimensional integral or sum. Moreover, $\lambda_j$ and $\nu_j$ are dependent on the iteration $k$ and need to be adjusted at every iteration to ensure the effectiveness of optimization.

Based on the structure of the optimal update policy, it is observed that as $\lambda_j \to 0$, $\pi^{i_h*}$ tends to be greedy while the policy becomes more exploratory when $\lambda_j$ increases. We note that $\lambda_j$ exhibits similarities to the temperature term utilized in maximum entropy reinforcement learning [33]. Fixed values of $\lambda$ have been demonstrated to yield reasonable outcomes in training [34, 35, 22]. In practical implementations, we have observed favorable results by using fixed $\lambda_j$ through hyperparameter sweeps. However continuous adaptation of $\nu_j$ during training is necessary to ensure cost constraint satisfaction. In this regard, we appeal to an intuitive heuristic based on primal-dual gradient methods [36] to determine the appropriate value of $\nu_j$. Recall that by strong duality, the optimal $\lambda_j^*$ and $\nu_j^*$ minimizes the dual function 5 which we will denote by $L(\pi^{i_h*}, \lambda_j, \nu_j)$. Then we can adopt gradient descent *w.r.t* $\nu_j$ to minimize $L(\pi^{i_h*}, \lambda_j, \nu_j)$ as follows:

**Corollary 2.** The derivative of $L(\pi^{i_h*}, \lambda_j, \nu_j)$ *w.r.t* $\nu_j$ is

$$\frac{\partial L(\pi^{i_h*}, \lambda_j, \nu_j)}{\partial \nu_j} = b_j^{i_h} - E_{s \sim \rho_{\pi_{\theta_k}}, a^{i_h} \sim \pi^{i_h*}(a|s)}[A_{j,\pi_{\theta_k}}^{i_h}(s, a^{i_h})]. \tag{9}$$

The proof is shown in Appendix D.

The last term in the gradient expression poses a challenge since $\pi^{i_h*}$ may not locate in parameterized policy space, leading direct evaluation of the term to be infeasible. Nevertheless, due to the proximity between $\pi_\theta^{i_h}$ and $\pi^{i_h*}$ enforced by the KL divergence constraint, it's reasonable to assume that $E_{s \sim \rho_{\pi_{\theta_k}}, a^{i_h} \sim \pi^{i_h*}}[A_{j,\pi_{\theta_k}}^{i_h}(s, a^{i_h})] \approx E_{s \sim \rho_{\pi_{\theta_k}}, a^{i_h} \sim \pi_{\theta_k}^{i_h}}[A_{j,\pi_{\theta_k}}^{i_h}(s, a^{i_h})] = 0$. In practice, we have observed that setting this term to zero yields favorable outcomes, which gives the update term as follows:

$$\nu_j \leftarrow \underset{\nu_j}{proj}[\nu_j - \alpha(c_j^{i_h} - J_j^{i_h}(\pi_{\theta_k}))], \tag{10}$$

where $\alpha$ is the step size to control the magnitude of the update. The projection operator $proj_{\nu_j}$ ensures that $\nu_j$ remains within the interval $[0, \nu_{max}]$, with $\nu_{max}$ chosen to prevent $\nu_j$ from being excessively large. In practical applications, the estimation of $J_j^{i_h}(\pi_{\theta_k})$ can be accomplished using Monte Carlo methods by leveraging trajectories collected from $\pi_{\theta_k}$. This approach aligns with the update rule employed in MACPO [24]. We recall that in 4, $\nu_j$ acts as a cost penalty term where increasing $\nu_j$ makes the likelihood of state-action pairs with higher costs being sampled by $\pi^{i_h*}$ diminish. Consequently, the update rule presented in 10 exhibits an intuitive characteristic: it raises $\nu_j$ if $J_j^{i_h}(\pi_{\theta_k}) > c_j^{i_h}$, indicating a violation of the cost constraint for $\pi_{\theta_k}$, and reduces $\nu_j$ otherwise. Using the proposed update rule, $\nu_j$ can be updated before updating the policy parameter $\pi^{i_h}$.

To be noted, our algorithm is a first-order method, which implies that the approximations made are accurate only around the initial condition (*i.e.*, $\pi_\theta^{i_h} = \pi_{\theta_k}^{i_h}$). To better enforce this condition, we have introduced a per-state acceptance indicator function $I(s_j) = \mathbf{1}_{D_{KL}(\pi_\theta^{i_h}, \pi_{\theta_k}^{i_h}) \leq \delta}$ to 27. This function helps in rejecting sampled states whose $D_{KL}(\pi_\theta^{i_h}, \pi_{\theta_k}^{i_h})[s]$ is too large, thereby improving the accuracy of the gradient update. The resulting sample gradient update equation is as follows:

$$\nabla_\theta L(\theta) = \frac{1}{N} \sum_{j=1}^{N} [\nabla_\theta D_{KL}(\pi_\theta^{i_h} || \pi_{\theta_k}^{i_h})[s_j] - \frac{\nabla_\theta \pi_\theta^{i_h}(a_j|s_j)}{\lambda \pi_{\theta_k}^{i_h}(a_j|s_j)}(\hat{\eta}_{\pi_{\theta_k}}(s, a^{i_h}) - \nu_j \hat{A}_{j,\pi_{\theta_k}}^{i_h}(s, a^{i_h}))]I(s_j), \tag{11}$$

where $N$ is the number of samples collected using policy $\pi_{\theta_k}$. The estimates of the advantages functions for the returns and costs, denoted as $\hat{\eta}_{\pi_{\theta_k}}$ and $\hat{A}_{j,\pi_{\theta_k}}^{i_h}$, are obtained from critic networks referring to MACPO. We estimate these advantages using Generalized Advantage Estimator (GAE) [37] and apply stochastic gradient descent using 11. During the training process, our algorithm employs the early stopping criterion to ensure that the updated policy satisfies the trust region constraint. Specifically, this criterion is defined as $\frac{1}{N} \sum_{j=1}^{N} D_{KL}(\pi_\theta^{i_h} || \pi_{\theta_k}^{i_h})[s_j] > \delta$. To update the value network, we minimize the Mean Square Error (MSE) between the output and target value. The procedure of our algorithm is presented in the Appendix E.

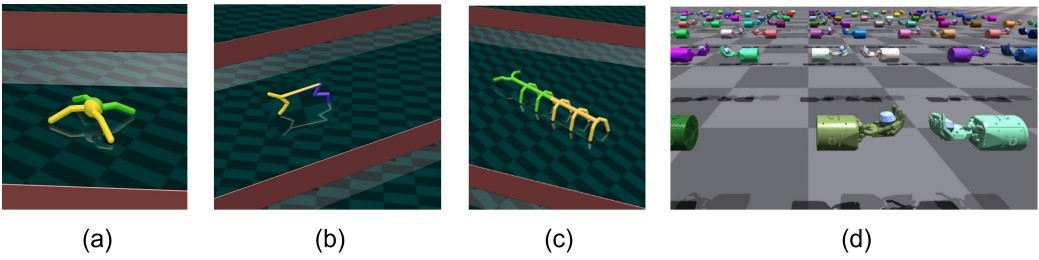

(a)          (b)          (c)          (d)

Figure 2: Example tasks in Safe MAMuJoCo and Safe MAIG. (a): Safe 2x4-Ant, (b): Safe 2x3-HalfCheetah, (c): Safe 2x3-ManyAgent Ant, (d): ShadowHandOver. In each of these tasks, the body parts of the robots are controlled by different agents. Agents collaborate to manipulate the robot while ensuring that the safety constraints are not violated.

## 5 Experiments

We evaluate the effectiveness of our algorithm on two benchmarks of safe MARL: Safe MAMuJoCo and Safe Multi-Agent Isaac Gym (MAIG). The former is a safety-aware modification of MAMuJoCo [38], where there exist obstacles in the environment. Meanwhile, Safe MAIG is developed on top of Issac Gym [39], a GPU-based platform for robotics tasks. Being an extension of DexterourHands [40], Safe MAIG requires agents to control the robot hands while optimizing both the reward and safety performance. We present some example tasks in Figure 5 and more details about the environments are introduced in the appendix.

While our primary aim is to propose a simpler alternative to MACPO, we also evaluate the performance of MAPPO-Lagrangian (MAPPO-L), which is put forward alongside MACPO [24]. This method adopts Lagrangian multipliers to solve optimization problems in MACPO. Without the requirements of repetitive computation of the Hessian matrix whose size grows quadratically with the dimension of the parameter vector, the Lagrangian method is also first-order and simple to implement. However, unlike MACPO and MAFOCOPS, whether this algorithm satisfies any worst-case constraint guarantees remains unknown. In addition, we compare the results of our algorithm with two standard MARL baseline algorithms, namely MAPPO [41]and HAPPO [18]. Both sets of experiments are carried out using the MACPO codebase and our experiments are conducted on GeForce RTX 3090 GPUS. More implementation details can be found in supplementary materials.

### 5.1 Performance on Safe MAMuJoCo

In this section, we select several experiment scenarios of Safe MAMuJoCo and execute each algorithm for 10 million samples per task. The cost thresholds are determined by taking 50% of the cost achieved by standard MARL algorithms after 1 million sample runs. To be noted, while most training parameters remain the same as those in the codebase, such as the learning rate and settings of optimizers, we adjusted some certain hyperparameters associated with the cost thresholds to make the algorithms best suit the experiment scenarios. It is noteworthy that the performance of MAPPO-L highly relies on the Lagrangian coefficient, rendering it more sensitive to the hyperparameters. In light of this observation, we adopt distinct hyperparameters for MAPPO-L in different categories of tasks. In contrast, the other two safe MARL algorithms are implemented with uniform hyperparameter settings across all experimental scenarios in this benchmark, indicating their potential efficacy.

Due to page limit, we present partial results of our experiments in Figures 3 and more results can be seen in Appendix G. The experimental results show that all the safety-aware algorithms are able to satisfy the safety constraints while our proposed MAFOCOPS manages to achieve the best overall performance across all tasks. Even when our method achieves similar performance to the other two algorithms in HalfCheetah scenarios, it still exhibits faster learning, demonstrating the advantages of our approach. For MAPPO-Lagrangian, we note that this baseline algorithm always achieves a similar performance as MAPPO, except in HalfCheetah scenarios where the cost threshold is significantly smaller compared to cost achieved by HAPPO and MAPPO. This may be due to that MAPPO-Lagrangian being built upon Lagrangian multiplier combined with standard MARL algorithms, leading to a performance more similar to safety-unaware MARL algorithms. In addition, from our discussion about MAPPO-L algorithm, we can know that it maintains a rather soft safety

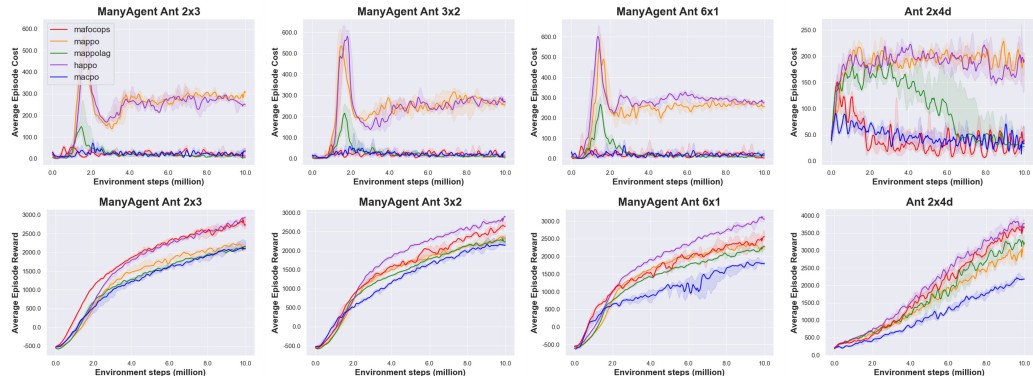

(a) ManyAgent Ant 2x3, ManyAgent Ant 3x2, ManyAgent Ant 6x1, Ant 2x4d

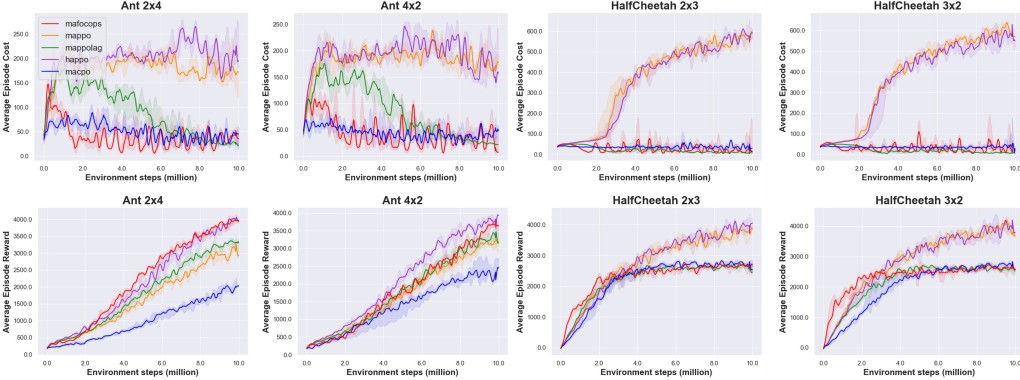

(b) Ant 2x4, Ant 4x2, HalfCheetah 2x3, HalfCheetah 3x2

Figure 3: Performance comparisons on tasks of Safe ManyAgent Ant, Ant, and HalfCheetah in terms of cost and reward. The safety constraint values for these presented tasks are set to be 25, 50 and 30, respectively.The solid line shows the median performance across 5 seeds and the shaded areas correspond to the 25-75% percentiles.

awareness, but the other two algorithms reaches safety via hard constraints. Therefore, MACPO and MAFOCOPS possess more promising properties, thus being the focus of our study.

Interestingly, our intuition suggests that a higher number of agents usually leads to increased complexity of the environment and hence worse performance. However, this phenomenon is not always observed. We assume that this is because, in the same type of map, the multi-agent system has to control the same robot, and the number of agents determines how the robot is partitioned. Fewer agents means that each agent has to control more parts of the robot, making the difficulty of the environment non-linearly dependent on the number of agents. Whereas, in scenarios of 6x1 ManyAgent Ant, we still note that as the number of agents increases, there exists a degradation in the performance of MACPO, which can attributed to the complexity of the computation worsening with the growth in the number of agents. As for MAFOCOPS, our first-order method shows stronger resistance to task complexity, showing the advantages of our algorithm when coping with multi-agent problems. What's more, we provide videos of the trained policies of both our algorithm and MACPO in the supplementary materials to intuitively demonstrate the benefits that our approach brings.

### 5.2 Performance on Safe MAIG

Apart from experiments in Safe MAMuJoCo, we also conduct experiments in the Safe MAIG benchmark. In this part of experiment, the cost thresholds are set as 25% of the cost obtained by standard MARL algorithms after running for one-tenth of the entire training process. The total training settings are the same as those in Safe MAMuJoCo, while some specific hyperparameter values are adjusted considering the great difference between these two experiment environments.

Experiment results in this benchmark are presented in Figure 4. Under this complicated environment,

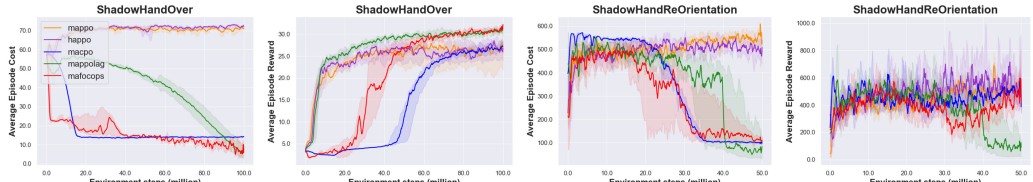

Figure 4: Performance comparisons on tasks of Safe MAIG. The safety constraint value for ShadowHandOver is set to be 15 and that for another task is set to be 110. The solid line shows the median performance across 5 seeds and the shaded areas correspond to the 25-75% percentiles.

it can be observed that the soft constraint algorithm, MAPPO-L, exhibits a delay in guaranteeing safety. Notably, in ShadowHandReOrientation task, it experiences a sudden drop in cost, which significantly impacts the reward. Conversely, although similar phenomenon occurs with MAFOCOPS, our method manages to optimize the reward in this condition and attains an acceptable performance. Although the reward of MACPO remains relatively stable, its final performance is not as favorable as our method. The different performance between MAPPO-L and the other two algorithms indicates that the property of upper bound violation is important in safe reinforcement learning, which also demonstrates the significance of our work.

### 5.3 Efficiency Analysis between MAFOCOPS and MACPO

As a first-order method, our MAFOCOPS not only eliminates the approximation errors when solving the optimization problems, leading to better performance, but also significantly reduces the computation cost. In this part, we evaluate the memory cost and frame per second (FPS), which is often adopted to measure the training efficiency of reinforcement learning algorithms, between MAFOCPS and MACPO. For simplicity, we adopt memory monitor tools to track memory utilization after 200000 samples and record the average FPS metric.

Due to the page limit, the results are shown in Appendix H of supplementary materials. From the results, it is evident that our algorithm obtains substantial improvement in computational efficiency and shows the ability to effectively save memory resources, especially when the number of agents grows. Consequently, we can infer that when confronted with tasks with multiple agents, second-order algorithms may be ill-suited due to their substantial computational costs. In contrast, our proposed method can successfully address such scenarios without succumbing to the computational burden.

### 5.4 Sensitivity Analysis

Based on the description provided in Section 4, it is apparent that the Lagrange multipliers, $\lambda_j$ and $\nu_{max}$, are crucial to the performance of our approach. In addition, analyzing the sensitivity of our algorithm to changes in the safety bound is also meaningful. Hence, we seek to investigate the sensitivity of our algorithm to these hyperparameters as well as the safety bound. In this way, we conduct some ablation studies using some scenarios of Safe MAMuJoCo and more details are presented in Appendix I.

Considering the intricacy of multi-agent environments, it is difficult to delineate the correlation between the performance of our method and the hyperparameters $\lambda_j$ and $\nu_{max}$. Nonetheless, choosing hyperparameters in proximity to the optimal values that we employed typically leads to favorable outcomes. Moreover, our approach's effectiveness is relatively insensitive to variations in these hyperparameter values. As an illustration, even setting $\nu_{max} = \infty$ does not significantly affect the reward achieved by our method, only resulting in an average 6.7% degradation. This finding indicates the robustness of our method, and further underscores its superiority over other approaches. What's more, from the sensitivity of our method to safety bound, we learn that although the reward performance of MAFOCOPS decreases with the stricter safety constraints, the algorithm's overall effectiveness remains unchanged across different safety levels.

## 6 Conclusion

In this paper, we introduce a novel method for training multi-agent systems while incorporating safety constraints. Building upon the problem formulation and sequential update scheme established in the

MACPO framework [24], our algorithm offers rigorous theoretical guarantees and relies exclusively on first-order optimization techniques, thereby ensuring simplicity in its implementation. Different from MACPO, which relies on second-order optimization to achieve parameterized policies, our work proposes a fundamentally different way to tackle the optimization problem, complemented by the comprehensive provision of derivatives. The empirical experiment results demonstrate the outstanding performance and computation efficiency of our algorithm, compared to the more intricate second-order method. To sum up, our paper offers a novel and distinct contribution to the realm of safe multi-agent reinforcement learning, as evidenced by the distinctions in methodology, proofs, and experimental results in comparison with previous work.

Because the benchmarks we adopt only return one cost for agents, the performance of our algorithm across multiple costs is not evaluated. In the future, we plan to test our approach in more environments and physical settings. Moreover, there exist a number of promising prospects for future research such as incorporating off-policy data to further enhance the training efficiency. Designing methods aimed at offline settings, which precludes interactions with environments, represents another valuable direction for study.

## Acknowledgments

This work is supported by National Key R&D Program of China under Contract 2022ZD0119802 as well as National Natural Science Foundation of China under Contract 61836011. What's more, it is funded by Collective Intelligence & Collaboration Laboratory (Open Fund Project No. QXZ23014101) and by Young Elite Scientists Sponsorship Program by CAST 2022QNRC001. Besides, this work is also supported by GPU cluster built by MCC Lab of Information Science and Technology Institution, USTC, and the Supercomputing Center of the USTC.

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

## A  Proof of the Optimization problem

According to the problem formulation, we give a definition of the "surrogate" cost, which aligns with what is employed in MACPO [24]:

**Definition.** Let $\boldsymbol{\pi}$ be a joint policy, and $\bar{\pi}^i$ be some other policy of agent $i$.Then for any of its costs of index $j \in 1, \cdots, m^i$, we define

$$L^i_{j,\boldsymbol{\pi}}(\bar{\pi}^i) = E_{s \sim \rho_{\boldsymbol{\pi}}, a^i \sim \bar{\pi}^i}[A^i_{j,\boldsymbol{\pi}}(s, a^i)].$$

In this way, consider $\boldsymbol{\pi}$ and $\bar{\boldsymbol{\pi}}$ be joint policies, $i \in \mathcal{N}$ be an agent and $j \in 1, \cdots, m^i$ be an index of one of its costs. From the proof of Theorem 1 in TRPO [42], (in particular, equations (41) $\sim$ (45)), applying it to joint policies $\boldsymbol{\pi}$ and $\bar{\boldsymbol{\pi}}$, we can conclude that

$$J^i_j(\bar{\boldsymbol{\pi}}) \leq J^i_j(\boldsymbol{\pi}) + E_{s \sim \rho_{\boldsymbol{\pi}}, \boldsymbol{a} \sim \bar{\boldsymbol{\pi}}}[A^i_{j,\boldsymbol{\pi}}(s, a^i)] + \frac{4\alpha^2 \gamma max_{s,a^i}|A^i_{j,\boldsymbol{\pi}}(s, a^i)|}{(1-\gamma)^2}, \tag{12}$$

where $\alpha = D^{max}_{TV}(\boldsymbol{\pi}, \bar{\boldsymbol{\pi}}) = max_s D_{TV}(\boldsymbol{\pi}(\cdot|s), \bar{\boldsymbol{\pi}}(\cdot|s))$. According to the definition of total variance divergence, defined by $D_{TV}(p||q) = \frac{1}{2}\sum_i |p_i - q_i|$, we can know that $D_{TV}(p||q) = D_{TV}(q||p)$. Using Pinsker's inequality $D_{TV}(p||q)^2 \leq \frac{D_{KL}(p||q)}{2}$ [43], we can change the order of policy in the divergence computation and obtain:

$$J^i_j(\bar{\boldsymbol{\pi}}) \leq J^i_j(\boldsymbol{\pi}) + E_{s \sim \rho_{\boldsymbol{\pi}}, \boldsymbol{a} \sim \bar{\boldsymbol{\pi}}}[A^i_{j,\boldsymbol{\pi}}(s, a^i)] + \frac{2\gamma max_{s,a^i}|A^i_{j,\boldsymbol{\pi}}(s, a^i)|}{(1-\gamma)^2}D^{max}_{KL}(\bar{\boldsymbol{\pi}}, \boldsymbol{\pi}). \tag{13}$$

It's to be noted that $E_{s \sim \rho_{\boldsymbol{\pi}}, \boldsymbol{a} \sim \bar{\boldsymbol{\pi}}}[A^i_{j,\boldsymbol{\pi}}(s, a^i)] = E_{s \sim \rho_{\boldsymbol{\pi}}, a^i \sim \bar{\pi}^i}[A^i_{j,\boldsymbol{\pi}}(s, a^i)]$ as the actions of other agents than $i$ do not change the value of the variable inside of the expectation. Furthermore, $D^{max}_{KL}(\bar{\boldsymbol{\pi}}, \boldsymbol{\pi}) = max_s D_{KL}(\bar{\boldsymbol{\pi}}(\cdot|s), \boldsymbol{\pi}(\cdot|s)) = max_s(\sum_{l=1}^n D_{KL}(\bar{\pi}^l(\cdot|s), \pi^l(\cdot|s))) \leq \sum_{l=1}^n max_s D_{KL}(\bar{\pi}^l(\cdot|s), \pi^l(\cdot|s))) = \sum_{l=1}^n D^{max}_{KL}(\bar{\pi}^l, \pi^l)$. Setting $\nu^i_j = \frac{2\gamma max_{s,a^i}|A^i_{j,\boldsymbol{\pi}}(s, a^i)|}{(1-\gamma)^2}$, we can finally obtain:

$$J^i_j(\bar{\boldsymbol{\pi}}) \leq J^i_j(\boldsymbol{\pi}) + L^i_{j,\boldsymbol{\pi}}(\bar{\pi}^i) + \nu^i_j \sum_{l=1}^n D^{max}_{KL}(\bar{\pi}^l, \pi^l) \tag{14}$$

The aforementioned equation is similar to Lemma 2 in MACPO, with the only difference being the order of policies in the Kullback-Leibler (KL) divergence term. However, this variation does not impact the subsequent derivations. To this end, we can establish the ultimate optimization problem presented in our work as follows:

$$\underset{\pi^{i_h}_\theta}{maximize}\, E_{s \sim \rho_{\boldsymbol{\pi}_{\boldsymbol{\theta}_k}}, a^{i_{1:h-1}} \sim \pi^{i_{1:h-1}}_{\theta_{k+1}}, a^{i_h} \sim \pi^{i_h}}[A^{i_h}_{\boldsymbol{\pi}_{\boldsymbol{\theta}_k}}(s, a^{i_{1:h-1}}, a^{i_h})] \tag{15}$$

$$s.t. J^{i_h}_j(\boldsymbol{\pi}_{\boldsymbol{\theta}_k}) + E_{s \sim \rho_{\boldsymbol{\pi}_{\boldsymbol{\theta}_k}}, a^{i_h} \sim \pi^{i_h}}[A^{i_h}_{j,\boldsymbol{\pi}_{\boldsymbol{\theta}_k}}(s, a^{i_h})] \leq c^{i_h}_j, \forall j \in 1, \cdots, m^{i_h} \tag{16}$$

$$\bar{D}_{KL}(\pi^{i_h}_\theta, \pi^{i_h}_{\theta_k}) \leq \delta. \tag{17}$$

where $\bar{D}_{KL}(\pi^{i_h}_\theta, \pi^{i_h}_{\theta_k}) \triangleq E_{s \sim \rho_{\boldsymbol{\pi}_{\boldsymbol{\theta}_k}}}[D_{KL}(\pi^{i_h}_{\theta_k}(\cdot|s), \pi^{i_h}_\theta(\cdot|s))]$.

## B  Proof of Theorem 1

We first demonstrate the optimization problem to be solved when finding optimization problem within nonparameterized policy space:

$$\underset{\pi^{i_h}}{maximize}\, E_{s \sim \rho_{\boldsymbol{\pi}_{\boldsymbol{\theta}_k}}, a^{i_{1:h-1}} \sim \pi^{i_{1:h-1}}_{\theta_{k+1}}, a^{i_h} \sim \pi^{i_h}}[A^{i_h}_{\boldsymbol{\pi}_{\boldsymbol{\theta}_k}}(s, a^{i_{1:h-1}}, a^{i_h})] \tag{18}$$

$$s.t. J^{i_h}_j(\boldsymbol{\pi}_{\boldsymbol{\theta}_k}) + E_{s \sim \rho_{\boldsymbol{\pi}_{\boldsymbol{\theta}_k}}, a^{i_h} \sim \pi^{i_h}}[A^{i_h}_{j,\boldsymbol{\pi}_{\boldsymbol{\theta}_k}}(s, a^{i_h})] \leq c^{i_h}_j, \forall j \in 1, \cdots, m^{i_h} \tag{19}$$

$$\bar{D}_{KL}(\pi^{i_h}, \pi^{i_h}_{\theta_k}) \leq \delta \tag{20}$$

$\boldsymbol{Proof.}$ We initiate our analysis by demonstrating the convexity of Problem (18-20) is convex *w.r.t.* $\pi^{i_h}$. Here we know that $\boldsymbol{\pi}_{\boldsymbol{\theta}_k}$ and $\theta^{i_{1:h-1}}_{k+1}$ are given and $J^{i_h}_j(\boldsymbol{\pi}_{\boldsymbol{\theta}_k})$ remains constant *w.r.t.* $\pi^{i_h}$. In this

way, $E_{s\sim\rho_{\pi_{\theta_k}},a^{i_{1:h-1}}\sim\pi_{\theta_{k+1}}^{i_{1:h-1}},a^{i_h}\sim\pi^{i_h}}[A_{\pi_{\theta_k}}^{i_h}(s,a^{i_{1:h-1}},a^{i_h})]$ and $E_{s\sim\rho_{\pi_{\theta_k}},a^{i_h}\sim\pi^{i_h}}[A_{j,\pi_{\theta_k}}^{i_h}(s,a^{i_h})]$ are similar so that we only need to consider the latter one. We can divide this formula like this: $E_{s\sim\rho_{\pi_{\theta_k}},a^{i_h}\sim\pi^{i_h}}[A_{j,\pi_{\theta_k}}^{i_h}(s,a^{i_h})] = \sum_s \rho_{\pi_{\theta_k}}(s)\sum_{a^{i_h}}\pi^{i_h}(a^{i_h}|s)A_{j,\pi_{\theta_k}}^{i_h}(s,a^{i_h})$, where $\rho_{\pi_{\theta_k}}(s)$ represents state visitation frequencies. We easily know that $\rho_{\pi_{\theta_k}}(s)$ is not affected by $\pi^{i_h}$. Similarly, for each action of agent $i_h$, the $A_{j,\pi_{\theta_k}}^{i_h}(s,a^{i_h})$ is also not relative to $\pi^{i_h}$. To this end, when $\pi_{\theta_k}$ and $\theta_{k+1}^{i_{1:h-1}}$ are given, $E_{s\sim\rho_{\pi_{\theta_k}},a^{i_h}\sim\pi^{i_h}}[A_{j,\pi_{\theta_k}}^{i_h}(s,a^{i_h})]$ is convex to $\pi^{i_h}$. Applying these analysis, the convexity of Equation 18 and 19 can be confirmed. Concerning constraint 20, it can be rewritten as $\sum_s \rho_{\pi_{\theta_k}}(s)D_{KL}(\pi^{i_h},\pi_{\theta_k}^{i_h})[s] \leq \delta$. Notably, KL divergence is convex *w.r.t.* its first argument, hence constraint 20 can be represented as a linear combination of convex functions, confirming its convexity as well. As $\pi_{\theta_k}^{i_h}$ fulfills constraint 19 and serves as an interior point within the set defined by constraint 20, therefore Slater's constraint qualification holds and strong duality holds.

Based on above discussion, we can solve for the optimal value for the problem (18 - 20) $p^*$ by solving the corresponding dual problem. We define $b_j^{i_h} = c_j^{i_h} - J_j^{i_h}(\pi_{\theta_k})$, then

$$L(\pi,\lambda_j,\nu_j) = \lambda_j\delta + \nu_j b_j^{i_h} + E_{s\sim\rho_{\pi_{\theta_k}}}[E_{a^{i_{1:h-1}}\sim\pi_{\theta_{k+1}}^{i_{1:h-1}},a^{i_h}\sim\pi^{i_h}}[A_{\pi_{\theta_k}}^{i_h}(s,a^{i_{1:h-1}},a^{i_h})]$$
$$- \nu_j E_{a^{i_h}\sim\pi^{i_h}}[A_{j,\pi_{\theta_k}}^{i_h}(s,a^{i_h})] - \lambda_j D_{KL}(\pi^{i_h}||\pi_{\theta_k}^{i_h})] \tag{21}$$

Therefore,

$$p^* = \max_{\pi_{i_h}\in\Pi}\min_{\lambda_j,\nu_j\geq 0} L(\pi,\lambda_j,\nu_j) = \min_{\lambda_j,\nu_j\geq 0}\max_{\pi_{i_h}\in\Pi} L(\pi,\lambda_j,\nu_j) \tag{22}$$

where we invoked strong duality in the second equality. According to the theory of convex optimization [44], if $\pi^{i_h*}, \lambda_j^*, \nu_j^*$ are optimal for 22, $\pi^{i_h*}$ is also optimal for Problem 18-20.

Consider the inner maximization problem in 22, we can decompose this problem into separate problems, one for each $s$.

$$\underset{\pi_{i_h}}{maximize}E_{a^{i_h}\sim\pi^{i_h}}[E_{a^{i_{1:h-1}}\sim\pi_{\theta_{k+1}}^{i_{1:h-1}}}[A_{\pi_{\theta_k}}^{i_h}(s,a^{i_{1:h-1}},a^{i_h})] - \nu_j A_{j,\pi_{\theta_k}}^{i_h}(s,a^{i_h})$$
$$- \lambda_j(log\pi^{i_h}(a|s) - log\pi_{\theta_k}^{i_h}(a|s))], \sum\pi^{i_h}(a|s) = 1 \tag{23}$$

As $E_{a^{i_{1:h-1}}\sim\pi_{\theta_{k+1}}^{i_{1:h-1}}}[A_{\pi_{\theta_k}}^{i_h}(s,a^{i_{1:h-1}},a^{i_h})]$ is irrelevant to $\pi^{i_h}$, we rename this term as $\eta_{\pi_{\theta_k}}^{i_h}(s,a^{i_h})$ for simplicity. This is clearly a convex optimization problem which can be solved using a simple Lagrangian argument. We can then get

$$G(\pi^{i_h}) = \sum_a \pi^{i_h}(a|s)[\eta_{\pi_{\theta_k}}^{i_h}(s,a^{i_h}) - \nu_j A_{j,\pi_{\theta_k}}^{i_h}(s,a^{i_h}) - \lambda_j(log\pi^{i_h}(a|s) - log\pi_{\theta_k}^{i_h}(a|s)) + \zeta] - \zeta \tag{24}$$

where $\zeta$ is the Lagrange multiplier associated with the constraint $\sum\pi^{i_h}(a|s) = 1$. Differentiating $G(\pi)$ w.r.t for some $a$:

$$\frac{\partial G}{\partial\pi^{i_h}(a|s)} = \eta_{\pi_{\theta_k}}^{i_h}(s,a^{i_h}) - \nu_j A_{j,\pi_{\theta_k}}^{i_h}(s,a^{i_h}) - \lambda_j(log\pi^{i_h}(a|s) - log\pi_{\theta_k}^{i_h}(a|s)) + \zeta \tag{25}$$

Set 25 to 0 and similar to FOCOPS, we can know

$$\pi^{i_h*}(a|s) = \frac{\pi_{\theta_k}^{i_h}(a|s)}{Z_{\lambda_j,\nu_j}(s)}exp\{\frac{1}{\lambda_j}(\eta_{\pi_{\theta_k}}(s,a^{i_h}) - \nu_j A_{j,\pi_{\theta_k}}^{i_h}(s,a^{i_h}))\} \tag{26}$$

where $Z_{\lambda_j,\nu_j}(s)$ is the partition function that ensures $\pi^{i_h*}$ to be a probability function, *i.e.*, $\sum_a \pi^{i_h*}(a|s) = 1$. Putting this $\pi^*$ back into equation 22, we can get

$$p^* = \min_{\lambda_j,\nu_j\geq 0} \lambda_j\delta + \nu_j b_j^{i_h} + E_{s\sim\rho_{\pi_{\theta_k}},a^{i_h}\sim\pi^{i_h*}}[\eta_{\pi_{\theta_k}}^{i_h}(s,a^{i_h}) - \nu_j A_{j,\pi_{\theta_k}}^{i_h}(s,a^{i_h}) - \lambda_j(log\pi^{i_h*}(a|s) - log\pi_{\theta_k}^{i_h}(a|s))]$$

$$= \min_{\lambda_j,\nu_j\geq 0} \lambda_j\delta + \nu_j b_j^{i_h} + E_{s\sim\rho_{\pi_{\theta_k}},a^{i_h}\sim\pi^{i_h*}}[\eta_{\pi_{\theta_k}}^{i_h}(s,a^{i_h}) - \nu_j A_{j,\pi_{\theta_k}}^{i_h}(s,a^{i_h}) - \lambda_j(log\pi_{\theta_k}^{i_h}(a|s) - logZ_{\lambda_j,\nu_j}$$

$$+ \frac{1}{\lambda_j}(\eta_{\pi_{\theta_k}}(s,a^{i_h}) - \nu_j A_{j,\pi_{\theta_k}}^{i_h}(s,a^{i_h})) - log\pi_{\theta_k}^{i_h}(a|s))]$$

$$= \min_{\lambda_j,\nu_j\geq 0} \lambda_j\delta + \nu_j b_j^{i_h} + \lambda_j E_{s\sim\rho_{\pi_{\theta_k}},a^{i_h}\sim\pi^{i_h*}}[logZ_{\lambda_j,\nu_j}(s)]$$

What's more, we give a simple description to show that for feasible policy $\pi_{\theta_k}$, the optimal policy update $\pi^{i_h*}$ has an upper bound for worst-case guarantee for cost constraint satisfaction. For agent $i_h$, according to Equation 14, after getting the optimal joint update policy for all agents, $J^i_j(\pi^*) \leq J^i_j(\pi_{\theta_k}) + L^i_{j,\pi_{\theta_k}}(\pi^{i_h*}) + \nu^{i_h}_j \sum^n_{l=1} D^{max}_{KL}(\pi^{l*}, \pi^l_{\theta_k})$ can be obtained. According to the definition of $L^i_{j,\pi}(\bar{\pi}^i)$ and the constraint 16 in the optimization problem, we can know that $J^i_j(\pi_{\theta_k}) + L^i_{j,\pi_{\theta_k}}(\pi^{i_h*}) \leq c^{i_h}_j$, thus leading to $J^i_j(\pi^*) \leq c^{i_h}_j + \nu^{i_h}_j \sum^n_{l=1} D^{max}_{KL}(\pi^{l*}, \pi^l_{\theta_k})$. In addition, we can know that the kl divergence between update policy and $\pi_{\theta_k}$ for each agent $l$ has an upper bound, which we call $\delta^l$. To this end, we achieve $J^i_j(\pi^*) \leq c^{i_h}_j + \frac{2\gamma max_{s,a^i}|A^i_{j,\pi}(s,a^i)|}{(1-\gamma)^2} \sum^n_{l=1} \delta^l$ , which is the upper bound for worst-case guarantee for cost constraint satisfaction. According to the result, we can know that with more agents, the upper bound for worst-case guarantee is higher, which means that optimization for more agents is more challenging, consistent with our intuition.

## C   Proof of Corollary 1

**Corollary 1.** The gradient of $L(\theta)$ takes the form

$$\nabla_\theta L(\theta) = E_{s \sim \rho_{\pi_{\theta_k}}}[\nabla_\theta D_{KL}(\pi^{i_h}_\theta||\pi^{i_h*})[s]] \tag{27}$$

where

$$\nabla_\theta D_{KL}(\pi^{i_h}_\theta||\pi^{i_h*})[s] = \nabla_\theta D_{KL}(\pi^{i_h}_\theta||\pi^{i_h}_{\theta_k}) - \frac{1}{\lambda_j} E_{a \sim \pi^{i_h}_{\theta_k}}[\frac{\nabla_\theta \pi^{i_h}_\theta(a|s)}{\pi^{i_h}_{\theta_k}(a|s)}(\eta_{\pi_{\theta_k}}(s,a^{i_h}) - \nu_j A^{i_h}_{j,\pi_{\theta_k}}(s,a^{i_h}))] \tag{28}$$

$\boldsymbol{Proof}$. Using the definition of KL divergence, we note that

$$D_{KL}(\pi^{i_h}_\theta||\pi^{i_h*}) = -\sum_a \pi^{i_h}_\theta(a|s)log\pi^{i_h*}(a|s) + \sum_a \pi^{i_h}_\theta(a|s)log\pi^{i_h}_\theta(a|s) = H(\pi^{i_h}_\theta, \pi^{i_h*})[s] - H(\pi^{i_h}_\theta)[s] \tag{29}$$

where $H(\pi^{i_h}_\theta)[s]$ is the entropy and $H(\pi^{i_h}_\theta, \pi^{i_h*})[s]$ is the cross-entropy. We expand the cross-entropy term which gives us:

$$H(\pi^{i_h}_\theta, \pi^{i_h*})[s] = -\sum_a \pi^{i_h}_\theta(a|s)log\pi^{i_h*}(a|s)$$

$$= -\sum_a \pi^{i_h}_\theta(a|s) * log(\frac{\pi^{i_h}_{\theta_k}(a|s)}{Z_{\lambda_j,\nu_j}} exp\{\frac{1}{\lambda_j}(\eta_{\pi_{\theta_k}}(s,a^{i_h}) - \nu_j A^{i_h}_{j,\pi_{\theta_k}}(s,a^{i_h}))\})$$

$$= -\sum_a \pi^{i_h}_\theta(a|s) * log\pi^{i_h}_{\theta_k}(a|s) + logZ_{\lambda_j,\nu_j}(s) - \frac{1}{\lambda_j}\sum_a \pi^{i_h}_\theta(a|s) * (\eta_{\pi_{\theta_k}}(s,a^{i_h}) - \nu_j A^{i_h}_{j,\pi_{\theta_k}}(s,a^{i_h}))$$

Then put this term back into Equation29:

$$D_{KL}(\pi^{i_h}_\theta||\pi^{i_h*})[s] = -\sum_a \pi^{i_h}_\theta(a|s) * log\pi^{i_h}_{\theta_k}(a|s) + \sum_a \pi^{i_h}_\theta(a|s)log\pi^{i_h}_\theta(a|s) + logZ_{\lambda_j,\nu_j}(s)$$

$$- \frac{1}{\lambda_j}\sum_a \pi^{i_h}_\theta(a|s) * (\eta_{\pi_{\theta_k}}(s,a^{i_h}) - \nu_j A^{i_h}_{j,\pi_{\theta_k}}(s,a^{i_h}))$$

$$= D_{KL}(\pi^{i_h}_\theta||\pi^{i_h}_{\theta_k}) + logZ_{\lambda_j,\nu_j}(s) - \frac{1}{\lambda_j} E_{a \sim \pi^{i_h}_{\theta_k}}[\frac{\pi^{i_h}_\theta(a|s)}{\pi^{i_h}_{\theta_k}(a|s)}(\eta_{\pi_{\theta_k}}(s,a^{i_h}) - \nu_j A^{i_h}_{j,\pi_{\theta_k}}(s,a^{i_h}))]$$

In this way, take the gradient on both sides and we can get:

$$\nabla_\theta D_{KL}(\pi^{i_h}_\theta||\pi^{i_h*})[s] = \nabla_\theta D_{KL}(\pi^{i_h}_\theta||\pi^{i_h}_{\theta_k}) - \frac{1}{\lambda_j} E_{a \sim \pi^{i_h}_{\theta_k}}[\frac{\nabla_\theta \pi^{i_h}_\theta(a|s)}{\pi^{i_h}_{\theta_k}(a|s)}(\eta_{\pi_{\theta_k}}(s,a^{i_h}) - \nu_j A^{i_h}_{j,\pi_{\theta_k}}(s,a^{i_h}))] \tag{30}$$

# D Proof of Corollary 2

**Corollary 2.** The derivative of $L(\pi^{i_h *}, \lambda_j, \nu_j)$ w.r.t $\nu_j$ is

$$\frac{\partial L(\pi^{i_h *}, \lambda_j, \nu_j)}{\partial \nu_j} = b_j^{i_h} - E_{s \sim \rho_{\pi_{\theta_k}}, a^{i_h} \sim \pi^{i_h *}(a|s)}[A_{j, \pi_{\theta_k}}^{i_h}(s, a^{i_h})] \tag{31}$$

$\boldsymbol{Proof.}$ From the definition of $L(\pi^{i_h *}, \lambda_j, \nu_j)$ and above discussion, we can know that

$$L(\pi^{i_h *}, \lambda_j, \nu_j) = \min_{\lambda_j, \nu_j \geq 0} \lambda_j \delta + \nu_j b_j^{i_h} + \lambda_j E_{s \sim \rho_{\pi_{\theta_k}}, a^{i_h} \sim \pi^{i_h *}}[log Z_{\lambda_j, \nu_j}(s)] \tag{32}$$

The first two terms is an affine function for $\nu_j$ we focus on the expectation in the last term.

$$\frac{\partial \pi^{i_h *}(a|s)}{\partial \nu_j} = \frac{\pi_{\theta_k}^{i_h}(a|s)}{Z_{\lambda_j, \nu_j}^2(s)} [Z_{\lambda_j, \nu_j}(s) * \frac{\partial exp(\frac{1}{\lambda_j}(\eta_{\pi_{\theta_k}}(s, a^{i_h}) - \nu_j A_{j, \pi_{\theta_k}}^{i_h}(s, a^{i_h})))}{\partial \nu_j}$$

$$- exp(\frac{1}{\lambda_j}(\eta_{\pi_{\theta_k}}(s, a^{i_h}) - \nu_j A_{j, \pi_{\theta_k}}^{i_h}(s, a^{i_h}))) * \frac{\partial Z_{\lambda_j, \nu_j}(s)}{\partial \nu_j}]$$

For simplicity, we record $exp(\frac{1}{\lambda_j}(\eta_{\pi_{\theta_k}}(s, a^{i_h}) - \nu_j A_{j, \pi_{\theta_k}}^{i_h}(s, a^{i_h})))$ as $e(x)$, so $\pi^{i_h *}(a|s) = \frac{\pi_{\theta_k}^{i_h}(a|s)}{Z_{\lambda_j, \nu_j}(s)} * e(x)$. In this way,

$$\frac{\partial \pi^{i_h *}(a|s)}{\partial \nu_j} = \frac{\pi_{\theta_k}^{i_h}(a|s)}{Z_{\lambda_j, \nu_j}^2(s)} [-\frac{A_{j, \pi_{\theta_k}}^{i_h}(s, a^{i_h})}{\lambda_j} Z_{\lambda_j, \nu_j}(s) e(x) - e(x) \frac{\partial Z_{\lambda_j, \nu_j}(s)}{\partial \nu_j}]$$

$$= -\frac{A_{j, \pi_{\theta_k}}^{i_h}(s, a^{i_h})}{\lambda_j} \pi^{i_h *}(a|s) - \pi^{i_h *}(a|s) \frac{\partial log Z_{\lambda_j, \nu_j}(s)}{\partial \nu_j} \tag{33}$$

Therefore, the derivative of the expectation in the last term of $L(\pi^{i_h *}, \lambda_j, \nu_j)$ can be written as:

$$\frac{\partial}{\partial \nu_j} E_{s \sim \rho_{\pi_{\theta_k}}, a^{i_h} \sim \pi^{i_h *}}[log Z_{\lambda_j, \nu_j}(s)] = E_{s \sim \rho_{\pi_{\theta_k}}, a^{i_h} \sim \pi_{\theta_k}^{i_h}}[\frac{\partial}{\partial \nu_j}(\frac{\pi^{i_h *}(a|s)}{\pi_{\theta_k}^{i_h}(a|s)} log Z_{\lambda_j, \nu_j}(s))]$$

$$= E_{s \sim \rho_{\pi_{\theta_k}}, a^{i_h} \sim \pi_{\theta_k}^{i_h}}[\frac{1}{\pi_{\theta_k}^{i_h}(a|s)}(\frac{\partial \pi^{i_h *}(a|s)}{\partial \nu_j} log Z_{\lambda_j, \nu_j}(s) + \pi^{i_h *}(a|s) \frac{\partial log Z_{\lambda_j, \nu_j}(s)}{\partial \nu_j})]$$

$$= E_{s \sim \rho_{\pi_{\theta_k}}, a^{i_h} \sim \pi_{\theta_k}^{i_h}}[\frac{\pi^{i_h *}(a|s)}{\pi_{\theta_k}^{i_h}(a|s)}(-\frac{A_{j, \pi_{\theta_k}}^{i_h}(s, a^{i_h})}{\lambda_j} log Z_{\lambda_j, \nu_j}(s) - \frac{\partial log Z_{\lambda_j, \nu_j}(s)}{\partial \nu_j} log Z_{\lambda_j, \nu_j}(s)) + \frac{\partial log Z_{\lambda_j, \nu_j}(s)}{\partial \nu_j})]$$

$$= E_{s \sim \rho_{\pi_{\theta_k}}, a^{i_h} \sim \pi^{i_h *}(a|s)}[-\frac{A_{j, \pi_{\theta_k}}^{i_h}(s, a^{i_h})}{\lambda_j} log Z_{\lambda_j, \nu_j}(s) - \frac{\partial log Z_{\lambda_j, \nu_j}(s)}{\partial \nu_j} log Z_{\lambda_j, \nu_j}(s)) + \frac{\partial log Z_{\lambda_j, \nu_j}(s)}{\partial \nu_j}]$$

In addition, according to the definition of $Z_{\lambda_j, \nu_j}$, we can get:

$$\frac{\partial Z_{\lambda_j, \nu_j}(s)}{\partial \nu_j} = \frac{\partial}{\partial \nu_j}(\sum_a \pi_{\theta_k}^{i_h}(a|s) exp\{\frac{1}{\lambda_j}(\eta_{\pi_{\theta_k}}(s, a^{i_h}) - \nu_j A_{j, \pi_{\theta_k}}^{i_h}(s, a^{i_h}))\}$$

$$= -\sum_a \pi_{\theta_k}^{i_h}(a|s) \frac{A_{j, \pi_{\theta_k}}^{i_h}(s, a^{i_h}))}{\lambda_j} e(x) = -\sum_a \frac{A_{j, \pi_{\theta_k}}^{i_h}(s, a^{i_h}))}{\lambda_j} \frac{\pi_{\theta_k}^{i_h}(a|s)}{Z_{\lambda_j, \nu_j}(s)} e(x) Z_{\lambda_j, \nu_j}(s)$$

$$= -\sum_a \frac{A_{j, \pi_{\theta_k}}^{i_h}(s, a^{i_h}))}{\lambda_j} \pi^{i_h *}(a|s) Z_{\lambda_j, \nu_j}(s)$$

$$= -\frac{Z_{\lambda_j, \nu_j}(s)}{\lambda_j} E_{a^{i_h} \sim \pi^{i_h *}}[A_{j, \pi_{\theta_k}}^{i_h}(s, a^{i_h}))]$$

What's more,

$$\frac{\partial log Z_{\lambda_j, \nu_j}(s)}{\partial \nu_j} = \frac{\partial Z_{\lambda_j, \nu_j}(s)}{\partial \nu_j} \frac{1}{Z_{\lambda_j, \nu_j}(s)} = -\frac{1}{\lambda_j} E_{a^{i_h} \sim \pi^{i_h *}}[A_{j, \pi_{\theta_k}}^{i_h}(s, a^{i_h}))] \tag{34}$$

Putting this result to above equation, we can get

$$\frac{\partial}{\partial \nu_j} E_{s \sim \rho_{\pi_{\theta_k}}, a^{i_h} \sim \pi^{i_h *}}[log Z_{\lambda_j, \nu_j}(s)]$$

$$= E_{s \sim \rho_{\pi_{\theta_k}}, a^{i_h} \sim \pi^{i_h *}(a|s)}\left[-\frac{A^{i_h}_{j,\pi_{\theta_k}}(s, a^{i_h})}{\lambda_j} log Z_{\lambda_j, \nu_j}(s) + \frac{A^{i_h}_{j,\pi_{\theta_k}}(s, a^{i_h})}{\lambda_j} log Z_{\lambda_j, \nu_j}(s) - \frac{A^{i_h}_{j,\pi_{\theta_k}}(s, a^{i_h})}{\lambda_j}\right]$$

$$= -\frac{1}{\lambda_j} E_{s \sim \rho_{\pi_{\theta_k}}, a^{i_h} \sim \pi^{i_h *}(a|s)}[A^{i_h}_{j,\pi_{\theta_k}}(s, a^{i_h})]$$

To sum up, the derivative of $\nu_j$ to function $L(\pi^{i_h *}, \lambda_j, \nu_j)$ can be written:

$$\frac{\partial L(\pi^{i_h *}, \lambda_j, \nu_j)}{\partial \nu_j} = b^{i_h}_j - E_{s \sim \rho_{\pi_{\theta_k}}, a^{i_h} \sim \pi^{i_h *}(a|s)}[A^{i_h}_{j,\pi_{\theta_k}}(s, a^{i_h})] \tag{35}$$

where $b^{i_h}_j = c^{i_h}_j - J^{i_h}_j(\pi_{\theta_k})$. In this way, we can update $\nu_j$ by $\nu_j \leftarrow proj_{\nu_j}[\nu_j - \alpha(c^{i_h}_j - J^{i_h}_j(\pi_{\theta_k}))]$

## E  Procedure of MAFOCOPS

In this section, we describe the procedure of our algorithm, outlined in Algorithm 1. To be noted, hyperparameters for each agent are identical throughout the algorithm.

---

**Algorithm 1** MAFOCOPS

---

**Require:** number of agents $n$, number of updates $K$, minibatch size $B$, temperature $\{\lambda_j\}_{1 \leq j \leq m^i}$, initial cost constraint parameter $\{\nu_j\}_{1 \leq j \leq m^i}$, cost constraint parameter bound $v_{max}$, learning rate for cost constraint parameter $\alpha_\nu$, trust region bound $\delta$, cost bound $b_j$
1: **Initialize**, policy networks $\{\pi^i_{\theta_0}, i \in \mathcal{N}\}$, global value network $\{\phi_0\}$ and cost value networks $\{\phi^i_{j,0}\}^{i \in \mathcal{N}}_{1 \leq j \leq m^i}$, replay buffer $\mathcal{B}$
2: **for** $k = 0, 1, \ldots$ **do**
3:     Generate trajectories $\tau \sim \pi_{\theta_k}$, save the data into the buffer and sample a batch of data;
4:     Estimate the C-returns $\hat{J}_C$ by averaging over the cost return for all episodes.
5:     Compute the advantage functions $\hat{A}_{\pi_{\theta_k}}(s, \boldsymbol{a})$ and $\hat{A}^i_{j,\pi_{\theta_k}}(s, a^i)$ using GAE;
6:     Draw a permutation $i_{1:n}$ of agents at random.
7:     Set $M^{i_1}(s, \boldsymbol{a}) = \hat{A}_{\pi_{\theta_k}}(s, \boldsymbol{a})$
8:     **for** agent $i_h = i_1, i_2, \cdots, i_n$ **do**
9:         Update $\nu_j$ by $\nu_j$ by $\nu_j \leftarrow proj_{\nu_j}[\nu_j - \alpha(c^{i_h}_j - \hat{J}^{i_h}_{C,j}(\pi_{\theta_k}))], \forall j = 1, \cdots, m^{i_h}$
10:        **for** $K$ epochs **do**
11:            **for** each minibatch data of size $B$ **do**
12:                Update value networks (and cost value networks analogously) by minimizing the MSE loss $\phi_{k+1} = argmin_\phi \sum_{t=0}^T (V_{\phi_k}(s_t) - \hat{R}_t)^2$, where $\hat{R}$ is the target return.
13:                Update policy network by the derived equation of $\nabla_\theta L(\theta)$ (Eq. 11), where $\hat{\eta}_{\pi_{\theta_k}}(s, a^{i_h})$ is estimated by $M^{i_{1:h}}(s, \boldsymbol{a})$.
14:            **end for**
15:            **if** $\bar{D}_{KL}(\pi^{i_h}, \pi^{i_h}_{\theta_k}) \geq \delta$ **then**
16:                Break
17:            **end if**
18:        **end for**
19:        Compute $M^{i_{1:h+1}}(s, \boldsymbol{a}) = \frac{\pi^{i_h}_{\theta^{i_h}_{k+1}}(a^{i_h}|o^{i_h})}{\pi^{i_h}_{\theta^{i_h}_k}(a^{i_h}|o^{i_h})} M^{i_{1:h}}(s, \boldsymbol{a})$, unless $h = n$
20:    **end for**
21: **end for**

---

## F  Experiment Environment Introduction

In this section, we introduce the environments that we adopt in the experiments.

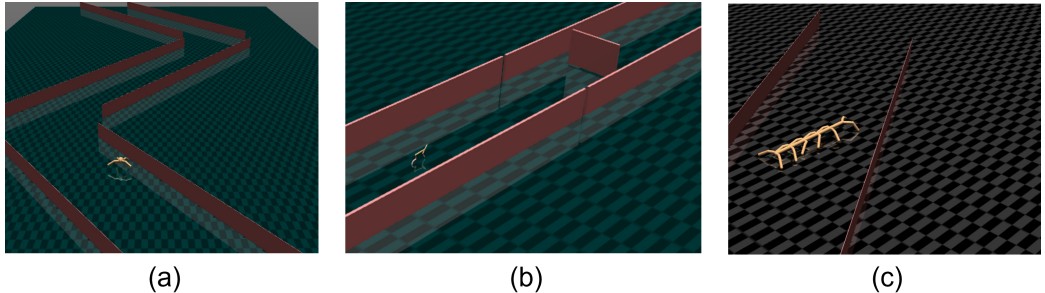

Figure 5: Specific tasks in Safe MAMuJoCo. (a): Ant Task: Ant 4x2 with three folding Jagged ($30°$) line walls, (b): HalfCheetah Task: HalfCheetah 2x3 with the moving obstacles, (c): ManyAgent Ant Task: ManyAgent Ant 2x3 inside one folding line walls (corridor width is 9 m).

## F.1 Safe MAMuJoCo

This environment is an extension of MAMuJoCo [38], maintaining the background environment, agents, physics simulator, and reward function. However, in the Safe MAMuJoCo setting, additional obstacles such as walls or pitfalls are introduced, and the environment emits cost with the increasing risk of an agent stumbling upon an obstacle. Here, we mainly introduce the scenarios that we employ in our work and present them in Figure 5.

**ManyAgent Ant task & Ant task** The corridor in the environment is bounded by two walls, with a width of 9 m for ManyAgent Ant and 10 m for Ant. The environment emits the cost of 1 for an agent, if the distance between the robot and the wall is less than 1.8 m, or when the robot topples over, which can be described as

$$c_t = \begin{cases} 1, & 0.2 \leq z_{torso,t+1} \leq 1.0, z_{rot} > -0.7, ||x_{torso,t+1} - x_{wall}||_2 \geq 1.8 \\ 0, & otherwise \end{cases}, \quad (36)$$

where $z_{torso,t+1}$ and $x_{torso,t+1}$ is the robot's torso's z-coordinate and x-coordinate at time $t+1$, $z_{rot}$ is the robot's rotation's z-coordinate and $x_{wall}$ denotes the x-coordinate of the wall.

**HalfCheetah task** In these maps, the HalfCheetah agents move inside a corridor (which constraints their movement, but does not induce costs). Concurrently, there are pitfalls within the corridor that also move. When an agent is too close to a pitfall, specifically when the distance between an agent and a pitfall is less than 9 m, a cost of 1 will be emitted.

$$c_t = \begin{cases} 1, & ||y_{torso,t+1} - y_{obstacle}||_2 \geq 9 \\ 0, & otherwise \end{cases}, \quad (37)$$

where the y-coordinate of the robot's torso is represented by $y_{torso,t+1}$ and $y_{obstacle}$ denotes the y-coordinate of the moving obstacles.

## F.2 Safe Multi-Agent Isaac Gym

This environment builds upon Issac Gym platform [39], renowned for its GPU-accelerated capabilities, and leverages the powerful Nvidia PhysX engine. Extending from the existing framework of DexterousHands [40], Safe MAIG requires agents to control the robot hands while optimizing both the reward and safety performance. Similarly, we also give an introduction of the specific scenarios in our experimental evaluations.

**ShadowHandOver** This task revolves around a dual-hand setup, with each hand occupying a fixed position. The primary objective entails the first hand, holding an object, navigating a suitable trajectory to transfer the item to the second hand while the second hand aims to acquire a successful grasp of the object. To be noted, this task incorporates safety constraints pertaining to the range of motion of one of the fingers on the first hand. Formally, the cost function can be expressed as follows:

$$c_t = \begin{cases} 1, & ||F_{a4,t+1}|| \geq 0.1 \\ 0, & otherwise \end{cases}, \quad (38)$$

where $F_{a4,t+1}$ is the first hand's fourth fingers's motion degree.

**ShadowHandReOrientation** Within the context of this task, both hands are equipped with two items. The fundamental objective for the agents is to execute rotational movements between these two items around each other and the safety constraints remain the same as Equation 38.

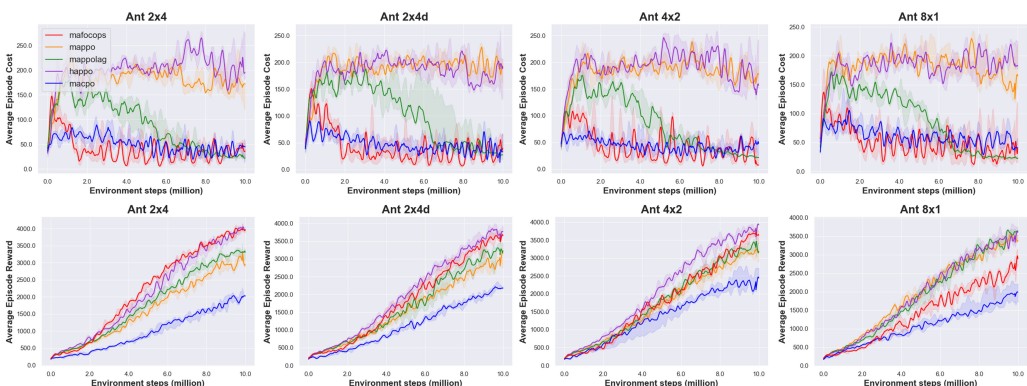

Figure 6: Performance comparisons on tasks of Ant 2x4, 2x4d, 4x2 and 8x1. The safety bound is 50, except for Ant 8x1 whose cost threshold is set as 70. The solid line shows the median performance across 5 seeds and the shaded areas correspond to the 25-75% percentiles.

## G    Performance on Safe MAMuJoCo

In this section, we present the comprehensive results of experiments in Safe MAMuJoCo environment in Figure 6-9. It can be observed that our proposed MAFOCOPS consistently demonstrates superior overall performance across all tasks. Even when our method achieves similar performance compared to the other two algorithms in HalfCheetah scenarios, it still exhibits faster learning, demonstrating the advantages of our approach. As is discussed in the Experiment section, MAPPO-L algorithm always achieves the similar performance as MAPPO, except in HalfCheetah scenarios where the cost threshold is significantly smaller compared to cost achieved by HAPPO and MAPPO. This may be due to that MAPPO-Lagrangian being built upon Lagrangian multiplier combined with standard MARL algorithms, leading to a performance more similar to safety-unaware MARL algorithms. Regarding the other two hard constraint algorithm, their performance would degrade with the increasing number of agents. However, MAFOCOPS consistently outperforms MACPO, proving the effectiveness of our method. What's more, we provide additional videos of the trained policies of both our algorithm and MACPO in supplementary materials.

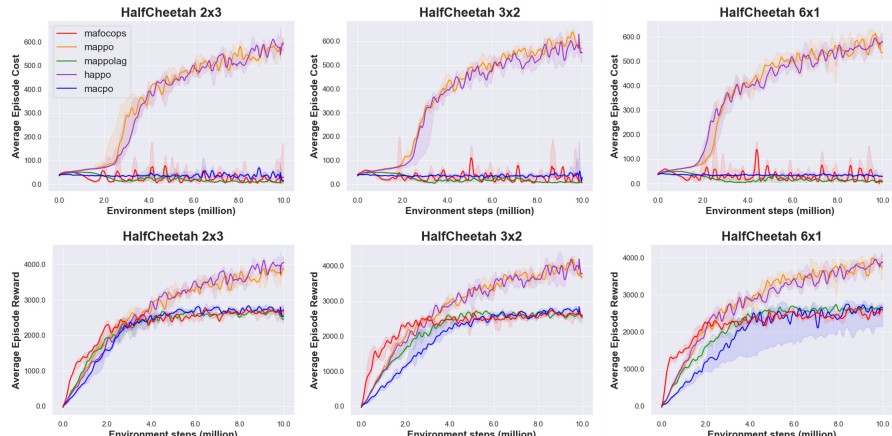

Figure 7: Performance comparisons on tasks of HalfCheetah 2x3, 3x2 and 6x1. The safety bound is 30. The solid line shows the median performance across 5 seeds and the shaded areas correspond to the 25-75% percentiles.

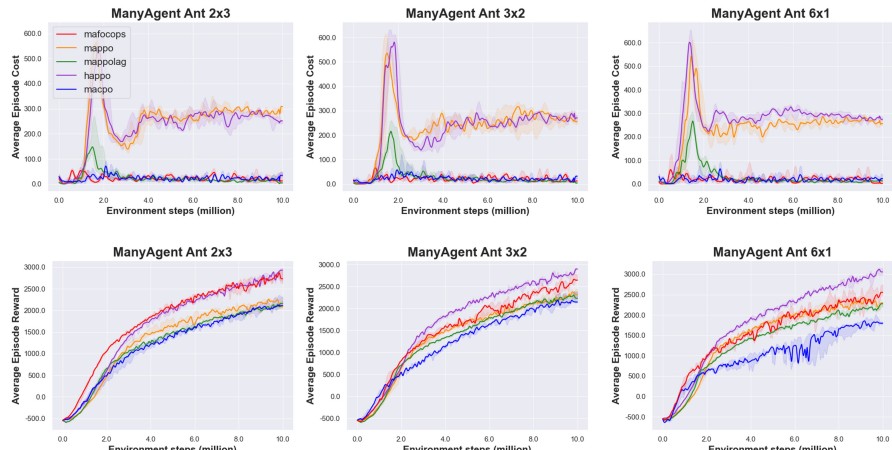

Figure 8: Performance comparisons on tasks of ManyAgent Ant 2x3, 3x2 and 6x1. The safety bound is 25. The solid line shows the median performance across 5 seeds and the shaded areas correspond to the 25-75% percentiles.

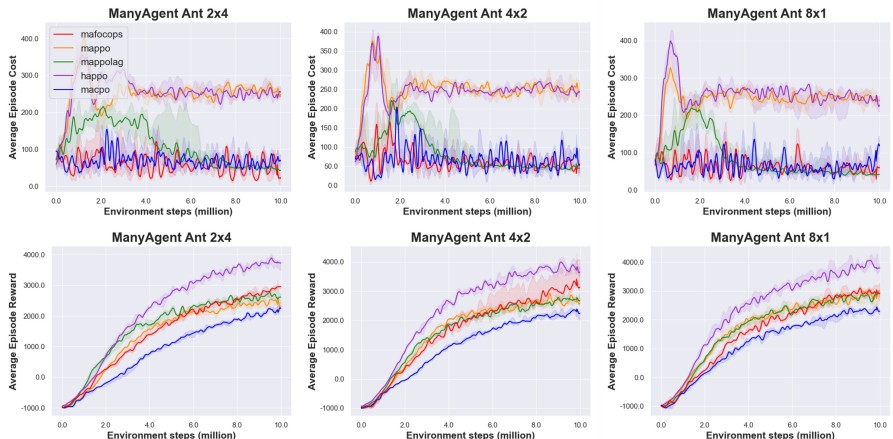

Figure 9: Performance comparisons on tasks of ManyAgent Ant 2x4, 4x2 and 8x1. The safety bound is 80. The solid line shows the median performance across 5 seeds and the shaded areas correspond to the 25-75% percentiles.

## H  Efficiency Analysis

In this section, we evaluate the training efficiency, which is measured by frame per second (FPS), and the memory cost between MACPO and our MAFOCOPS. To be specific, we record the time and samples spent for each update to calculate average FPS and employ memory monitor tools to track memory utilization after 200000 samples. To ensure a fair comparison, both algorithms are executed on the same GPU device, thereby minimizing the influence of other variables. The results obtained from these evaluations are presented in Table 1 and Table 2 with a precision of two decimal places. Based on the obtained results, it is evident that an increase in the number of agents leads to a noticeable escalation in computational cost for MACPO. Whereas, our algorithm showcases substantial improvement in computational efficiency and demonstrates the ability to effectively conserve memory resources, especially in scenarios involving a larger number of agents.

## I  Sensitivity Analysis

We test the sensitivity of our algorithm to hyperparameters, *i.e.,* $\lambda_j$ and $\nu_{max}$, as well as the safety bound. To be noted, because the benchmarks that we adopt only involve a single cost, we only need to set one value for $\lambda_j$ and $\nu_{max}$. In future works, we may explore the performance of our method in

| Scenarios | Ant Task | | | | HalfCheetah Task | | |
|---|---|---|---|---|---|---|---|
| Config
FPS | 2x4d | 2x4 | 4x2 | 8x1 | 2x3 | 3x2 | 6x1 |
| MACPO | 231 | 218 | 130 | 73 | 298 | 192 | 106 |
| MAFOCOPS | 322 | 270 | 160 | 115 | 340 | 229 | 162 |
| Improvement(%) | **39.39** | **23.85** | **23.08** | **57.53** | **14.09** | **19.27** | **52.83** |
| Scenarios | ManyAgent Ant Task | | | | | | |
| Config
FPS | 2x3 | 3x2 | 6x1 | – | 2x4 | 4x2 | 8x1 |
| MACPO | 244 | 167 | 98 | – | 232 | 135 | 73 |
| MAFOCOPS | 271 | 249 | 149 | – | 253 | 193 | 115 |
| Improvement(%) | **11.07** | **49.10** | **52.04** | – | **9.05** | **42.96** | **57.53** |

Table 1: Average FPS between MACPO and MAFOCOPS and the bold results demonstrate the improvement brings by our algorithm.

| Scenarios | Ant Task | | | | HalfCheeath Taks | | |
|---|---|---|---|---|---|---|---|
| Config
Memory (MiB) | 2x4d | 2x4 | 4x2 | 8x1 | 2x3 | 3x2 | 6x1 |
| MACPO | 18.85 | 23.60 | 31.24 | 66.25 | 16.54 | 30.20 | 52.08 |
| MAFOCOPS | 18.97 | 21.82 | 24.23 | 56.99 | 19.34 | 27.26 | 39.15 |
| Saved Memory | -0.12 | **1.77** | **7.01** | **9.26** | -2.80 | **2.93** | **12.93** |
| Scenarios | ManyAgent Ant Task | | | | | | |
| Config
Memory (MiB) | 2x3 | 3x2 | 6x1 | – | 2x4 | 4x2 | 8x1 |
| MACPO | 25.32 | 32.64 | 55.27 | – | 27.31 | 38.90 | 65.02 |
| MAFOCOPS | 24.45 | 30.88 | 44.38 | – | 24.62 | 34.73 | 60.71 |
| Saved Memory | **0.87** | **1.76** | **10.89** | – | **2.69** | **4.17** | **4.31** |

Table 2: Memory cost of MACPO and MAFOCOPS and the bold results demonstrate the memory saved by our algorithm.

environments with multiple costs. We choose several scenarios in Safe MAMuJoCo to conduct the ablation studies.

The sensitivity to the hyperparameters is evaluated across several different values for $\lambda_j$ and $\nu_{max}$ while keeping all other parameters fixed. For ease of comparison, we normalized the results based on the return and cost achieved by [18], namely if our method yields a return of $x$ and HAPPO achieves a return of $y$, the normalized result is reported as $\frac{x}{y}$. The results report the final performance of the models after training for 10 million steps and are showcased in Table 3 and Table 4 with a precision of three decimal places. Given the complexity inherent in multi-agent environments, it is difficult to delineate the correlation between the performance of our method and the hyperparameters $\lambda_j$ and $\nu_{max}$. It can be observed that different scenarios have different sensitivity to these hyperparameters from the results. Overall, the reward achieved under different settings is relatively insensitive as even setting these parameters across a broad range only leads to an average degradation of less than 10%. On the other hand, the cost may be more sensitive to these parameters, highlighting the inherent challenges in ensuring safety guarantees in safe multi-agent reinforcement learning to some degree.

Furthermore, we select some maps to examine the sensitivity of our algorithm to the safety bound. To be mentioned, hyperparameters in this experiment remain unchanged as the main experiments. From the results depicted in Figure 10, we can see that setting the safety bound too high could lead to increased oscillations in cost performance, although it may yield better reward performance. This observation is reasonable since higher safety bounds may allow agents to explore actions with potentially higher returns but less safety. If the safety bound is low enough, the agents will take actions that are definitely safe, leading to less vibration. Whereas, from a global perspective, the effectiveness of our algorithm remains consistent across these different safety levels. In general, we

| | Ant 2x4 | | HalfCheetah 2x3 | | ManyAgent Ant 2x3 | | ManyAgent Ant 2x4 | | All envs | |
|---|---|---|---|---|---|---|---|---|---|---|
| $\lambda$ | Reward | Cost | Reward | Cost | Reward | Cost | Reward | Cost | Reward | Cost |
| 1 | 0.913 | 0.212 | 0.599 | 0.049 | 0.889 | 0.137 | 0.781 | 0.082 | 0.796 | 0.120 |
| 2 | 0.975 | 0.188 | 0.658 | 0.056 | 0.946 | 0.131 | 0.882 | 0.212 | 0.865 | 0.147 |
| 2.2 | 0.964 | 0.091 | 0.668 | 0.049 | 0.947 | 0.090 | 0.802 | 0.166 | 0.845 | 0.099 |
| 3 | 0.983 | 0.183 | 0.699 | 0.073 | 0.958 | 0.059 | 0.879 | 0.178 | 0.880 | 0.123 |
| 5 | 1.004 | 0.113 | 0.694 | 0.078 | 0.871 | 0.070 | 0.784 | 0.267 | 0.838 | 0.132 |

Table 3: Performance of MAFOCOPS for different $\lambda$ and the "all envs" column presents the averaged performance across these four scenarios.

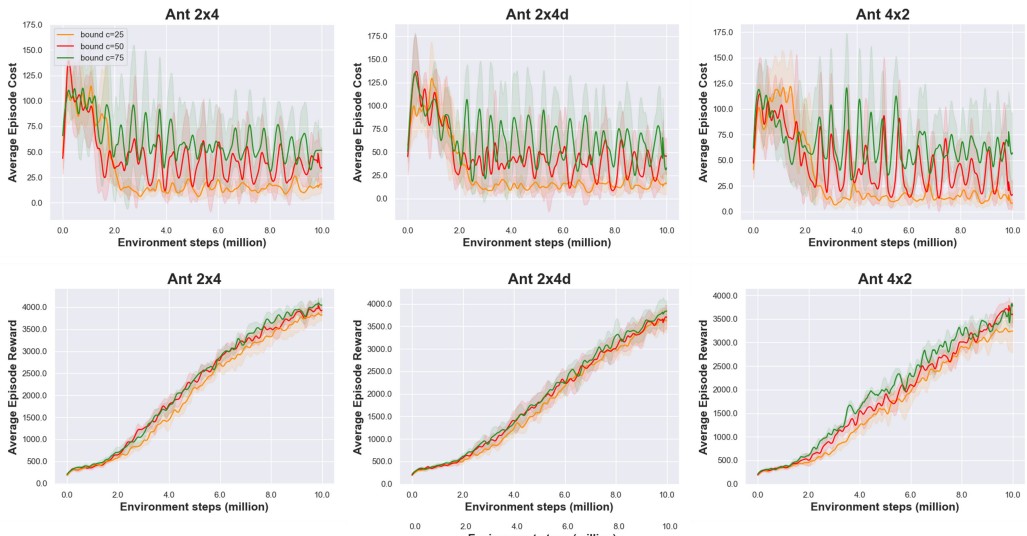

Figure 10: Performance comparisons on Ant 2x4, 2x4d, 4x2 with different safety bound.

need to strike a balance between ensuring safety and achieving a good reward performance when applying safe RL algorithms.

| | Ant 2x4 | | HalfCheetah 2x3 | | ManyAgent Ant 2x3 | | ManyAgent Ant 2x4 | | All envs | |
|---|---|---|---|---|---|---|---|---|---|---|
| $\nu_{max}$ | Reward | Cost | Reward | Cost | Reward | Cost | Reward | Cost | Reward | Cost |
| 1 | 1.042 | 0.074 | 0.680 | 0.029 | 0.981 | 0.067 | 0.859 | 0.328 | 0.891 | 0.125 |
| 1.3 | 0.964 | 0.091 | 0.668 | 0.049 | 0.947 | 0.090 | 0.802 | 0.166 | 0.845 | 0.099 |
| 2 | 0.908 | 0.222 | 0.627 | 0.040 | 0.962 | 0.137 | 0.910 | 0.206 | 0.852 | 0.151 |
| 3 | 0.923 | 0.097 | 0.579 | 0.042 | 0.936 | 0.077 | 0.844 | 0.123 | 0.821 | 0.085 |
| 5 | 0.793 | 0.175 | 0.606 | 0.063 | 1.013 | 0.102 | 0.800 | 0.256 | 0.803 | 0.149 |
| $\infty$ | 0.873 | 0.209 | 0.588 | 0.048 | 0.959 | 0.100 | 0.749 | 0.146 | 0.792 | 0.126 |

Table 4: Performance of MAFOCOPS for different $\nu_{max}$ and the "all envs" column presents the averaged performance across these four scenarios.

## J  Details of Settings for Experiments

The majority of settings have been described in detail in the Experiments section; however, we provide some additional information here. As our implementation is based on the codebase provided by MACPO [24], and thus most hyperparameters remain consistent with their original values. For MAFOCOPS, the Lagrange multipliers, namely $\lambda$ and $\nu_{max}$, we utilize are 2.2 and 1.3, respectively, which can found in Table 3 and 4. For the other two safe MARL algorithms, MACPO and MAPPO-L, we modify the relevant hyperparameters to ensure their compatibility with the safety bound As mentioned in the Experiments section, for the two benchmarks, we adopt distinct hyperparameters

for MAPPO-L in different categories of tasks, as the safety bound is relative to the cost achieved by standard MARL algorithms. However, MAFOCOPS and MACPO both use unchanged parameters, indicating the robustness of these two methods. We present the specific hyperparameters that we use in our experiments in Table 5 (as most parameters are unchanged, we only report the changed ones or unique parameters in our algorithm).

| Safe MAMuJoCo | MACPO | MAPPO-L | MAFOCOPS |
|---|---|---|---|
| kl-threshold | 0.008 | / | 0.0125 |
| lambda lagr | / | $[0.38^a, 0.46^b, 0.59^c, 0.52^d]$ | / |
| $\lambda$ | / | / | 2.2 |
| $\nu_{max}$ | / | / | 1.3 |
| $\nu$ lr | / | / | 0.00005 |
| fraction coef | 0.3 | / | / |
| minibatch size | / | / | 256 |
| update numbers | / | / | 5 |
| Safe MAIG | MACPO | MAPPO-L | MAFOCOPS |
| kl-threshold | 0.009 | / | 0.01 |
| lambda lagr | / | $[0.14^a, 0.68^b]$ | / |
| lagrangian coef rate | / | $[1e-7^a, 9e-7^b]$ | / |
| $\lambda$ | / | / | 2 |
| $\nu_{max}$ | / | / | 1.4 |
| $\nu$ lr | / | / | 0.001 |
| fraction coef | 0.26 | / | / |
| minibatch size | / | / | 8192 |
| update numbers | / | / | 3 |

Table 5: Different hyperparameters used for MACPO, MAPPO-L and MAFOCOPS. As MAPPO-L employs different hyperparameters, the changed ones are represented in the list. In Safe MAMuJoCo domains, a means Ant tasks, b corresponds to HalfCheetah tasks, c represents ManyAgent Ant 2x3 tasks and d represents denotes ManyAgent Ant 2x4 tasks. In Safe MAIG domains, a represents ShadowHandOver task and b denotes ShadowHandReOrientation task.

