# OpenReview forum: "Multi-Agent First Order Constrained Optimization in Policy Space"
_NeurIPS.cc/2023/Conference — NeurIPS 2023 poster_

### Official Review · Reviewer_VTR2 · 2023-07-04

**Soundness:** 4 excellent
**Presentation:** 4 excellent
**Contribution:** 3 good
**Rating:** 7
**Confidence:** 3

**Summary:**

The paper introduces a fresh approach to tackle the problem of safe Multi-Agent Reinforcement Learning (MARL) within a fully-cooperative multi-agent environment where all agents share a common reward function. The authors propose a novel algorithm known as Multi-Agent First Order Constrained Optimization in Policy Space (MAFOCOPS). This algorithm aims to maximize the expected total reward while ensuring each agent adheres to its safety constraint. The authors claim that MAFOCOPS is easy to implement and provides an approximate upper bound for worst-case constraint violation. To demonstrate its effectiveness, the paper conducts experiments on two safe MARL benchmarks, Safe MAMuJoCo and Safe MAIG. The results show that MAFOCOPS outperforms MACPO.

**Strengths:**

1.	The paper introduces a unique approach to multi-agent reinforcement learning, focusing on a fully-cooperative setting. It provides a new perspective on the problem by considering the influence of an agent's action on the total costs, even if it doesn't directly impact the costs of other agents. This approach captures the realistic multi-agent interactions in the real world, such as the disruption in traffic flow caused by a car running a red light.

2.	The paper is well-structured and provides a clear explanation of the proposed method. It also includes a proof in the appendix, demonstrating the mathematical rigor of the work. The paper also compares its method with other algorithms, showing that it maintains a soft safety awareness, unlike other algorithms that reach safety via hard constraints.

3.	The approach is significant as it provides a new perspective on multi-agent reinforcement learning, considering the influence of an agent's action on the total costs. The empirical results demonstrate the outstanding performance and computational efficiency of the proposed method, compared to more intricate second-order methods. The paper also suggests future work, planning to test the approach in more environments and physical settings.


**Weaknesses:**

The paper does not evaluate the performance of the algorithm across multiple costs. The benchmarks adopted only return one cost for agents, which limits the comprehensiveness of the evaluation. This could potentially mask some weaknesses or limitations of the proposed method when dealing with multiple costs.

**Questions:**

1.	The paper acknowledges that solving the dual problems presented in the optimization process is computationally challenging when dealing with large state/action spaces. Calculating the partition function often involves evaluating a high-dimensional integral or sum, which can be computationally intensive. Moreover, the parameters λj and νj are dependent on the iteration k and need to be adjusted at every iteration to ensure the effectiveness of optimization. This adds to the complexity of the implementation. It would be beneficial if the authors could provide more details on how they plan to address these challenges in practical implementations. Are there any strategies or methods that could be used to reduce the computational complexity?

2.	The paper mentions that the proposed method only employs first-order approximations, making it straightforward to implement. However, could the use of only first-order approximations limit the accuracy or effectiveness of the method in certain scenarios?

3.	The paper mentions that the proposed method has an approximate upper bound for worst-case constraint violation. Could the authors elaborate more on what this upper bound is and how it was determined?

4.	The paper discusses the use of a two-step process to solve the optimization problem. Could there be potential issues with this approach, such as the possibility of getting stuck in local optima?


**Limitations:**

Based on the information extracted from the paper, the authors have acknowledged some limitations of their work. They mention that due to the benchmarks they used, the performance of their algorithm across multiple costs was not evaluated. They also express their intention to test their approach in more environments and physical settings in the future.

---

> ### Author Rebuttal · Authors · 2023-08-09
>
> *Q1*: It would be beneficial if the authors could provide more details on how they plan to address computational complexity in practice. Are there any strategies or methods that could be used to reduce the computational complexity?
>
> *A1*: We introduce the implementation of these two hyperparameters in the Section 4.3. In fact, we note that $\lambda_j$ is similar to the temperature term utilized in maximum entropy RL [1].  Previous studies have shown that fixed values of $\lambda$ can yield reasonable outcomes [2]. Therefore, in practice, we adopt fixed $\lambda_j$ through grid search, allowing us to strike a balance between computational efficiency and model performance.
>
> As for $\nu_j$, we derive Corollary 2, which enables us to apply gradient descent method to optimize this parameter. In practice, we use Equation (10) to update $\nu_j$. The equation exhibits an intuitive characteristic: it raises $\nu_j$ if $J_j^{i_h}(\boldsymbol{\pi_{\theta_k}}) > c_j^{i_h}$, indicating a violation of the cost constraint, and reduces $\nu_j$ otherwise. Aligning with the update rule employed in MACPO to estimate $J_j^{i_h}(\boldsymbol{\pi_{\theta_k}})$, $\nu_j$ can be updated during training, which is shown in the Procedure of our algorithm in the Appendix. We think using these strategies can greatly reduce the computational complexity. The efficiency analysis can also illustrate the efficiency of these strategies in some way.
>
> *Q2*: The paper mentions that the proposed method only employs first-order approximations, making it straightforward to implement. However, could the use of only first-order approximations limit the accuracy or effectiveness of the method in certain scenarios?
>
> *A2*: We acknowledge that first-order approximations may introduce a trade-off between efficiency and accuracy. However, we would like to emphasize that despite employing first-order approximations, our method exhibits superior empirical performance compared to baseline algorithms.  This indicates that the approximations are not detrimental to the overall effectiveness of our approach. Meanwhile, second-order methods will introduce many approximation errors, so whether they are more accurate than first-order methods is hard to determine. Moreover, one of the significant strengths of our method is its obvious improved efficiency compared to MACPO, where the use of first-order approximations is crucial. Even if in some certain scenarios using only first-order approximations may limit the accuracy, we believe that our approach can strike a well-balanced compromise between efficiency and accuracy.
>
> *Q3*: The paper mentions that the proposed method has an approximate upper bound for worst-case constraint violation. Could the authors elaborate more on what this upper bound is and how it was determined?
>
> *A3*: Due to the page limit, the approximate upper bound is given in the Appendix. Here we give a brief description due to character limit. For each agent $i$, after getting the optimal joint update policy for all agents, $J_j^i(\boldsymbol{\pi^*}) \leq J_j^i(\boldsymbol{\pi_{\theta_k}}) + L_{j, \boldsymbol{\pi_{\theta_k}}}^i(\pi^{i_h*}) + \nu_j^{i_h} \sum_{l=1}^n D_{KL}^{max}(\pi^{l*}, \pi_{\theta_k}^l)$ can be obtained, where $L_{j, \boldsymbol{\pi}}^i(\bar{\pi}^i) = E_{s \sim \rho_{\boldsymbol{\pi}}, a^i \sim \bar{\pi}^i}[A_{j,\boldsymbol{\pi}}^i(s,a^i)]$. Considering the constraint in the optimization problem, we can know that $J_j^i(\boldsymbol{\pi^*}) \leq c_j^{i_h}+ \nu_j^{i_h} \sum_{l=1}^n D_{KL}^{max}(\pi^{l*}, \pi_{\theta_k}^l)$. Besides,  the kl divergence for each $l$ has an upper bound, which we call $\delta^l$. To this end, we achieve $J_j^i(\boldsymbol{\pi^*}) \leq c_j^{i_h} + \frac{2 \gamma max_{s,a^i}|A^i_{j.\boldsymbol{\pi}}(s,a^i)|}{(1-\gamma)^2} \sum_{l=1}^n \delta^l$ , which is the upper bound for worst-case constraint violation. This bound indicates that with more agents, the optimization becomes more challenging, aligning with our intuition.
>
> *Q4*: The paper discusses the use of a two-step process to solve the optimization problem. Could there be potential issues with this approach, such as the possibility of getting stuck in local optima?
>
> *A4*: We appreciate your comments and understand your concern. In fact, the idea of solving problems in nonparametric space and then projecting back into the parameter space has been successfully applied to tackle various challenging problems. For instance, Abdolmaleki et al. [3] takes the view of ''inference view'' of policy search and attempts to find the desired policy via the EM algorithm. Montgomery et al. [4] develop a guided policy update method and provide theoretical guarantees on the bounded error in the projection step. In addition, Supervised Policy Update (SPU) [5] solves a constrained optimization problem in the nonparameteric policy space and then converts the optimal policy to a parameterized one using supervised regression. These previous works provide theoretical foundations for our methodology. Furthermore, the performance of our algorithm in extensive experiments supports the validity of our approach, showing that the two-step process does not lead our method to getting stuck in local optima or other significant issues.
>
> [1] B. D. Ziebart et al., “Maximum entropy inverse reinforcement learning.” in Aaai, vol. 8. 2008, pp. 1433–1.
>
> [2] T. Haarnoja et al, “Soft actor-critic: Off-policy maximum entropy deep reinforcement learning with a stochastic actor,” in International Conference on Machine Learning, 2018, pp. 1861–1870.
>
> [3]A. Abdolmaleki et al, “Maximum a posteriori policy optimisation,” in International Conference on Learning Representations, 2018.
>
> [4] W. H. Montgomery and S. Levine, “Guided policy search via approximate mirror descent,” Advances in Neural Information Processing Systems, vol. 29, 2016.
>
> [5] Q. Vuong et al, “Supervised policy update for deep reinforcement learning,” in International Conference on Learning Representations, 2018.

---

### Official Review · Reviewer_eMiG · 2023-07-05

**Soundness:** 2 fair
**Presentation:** 2 fair
**Contribution:** 2 fair
**Rating:** 3
**Confidence:** 3

**Summary:**

This paper proposes a first-order method to solve the safety-constrained multi-agent policy optimization problem. The method is evaluated on two benchmarks and shows better performance over the baselines.

**Strengths:**

- The studied problem is important and might interest the community.
-  The writing is easy to follow.

**Weaknesses:**

- My main concern is about the correctness of the derivation in the proposed method. The proof of *Throrem 1* says both *Equation (7)*, the objective and *Equation (8)*, the cost constraint, are **linear** *w.r.t.* $\pi^{i_h}$, which looks strange to me. The equations involve advantage estimation which definitely has a non-linear relationship with $\pi^{i_h}$. Why would it be linear?


- When solving the optimization problem, in *Corollary 2*, the expectation of the advantage over the optimal policy $\pi^{i_h\ast}$ is assumed to be zero. The advantage means the performance improvement of the optimal policy. How can it be reasonable to assume it to be zero?

**Questions:**

- Please see the above *Weakness* section.

**Limitations:**

- The correctness of the proposed method seems skeptical. Several important steps in the derivation are confusing or taken as granted without sufficient argument.

---

> ### Author Rebuttal · Authors · 2023-08-09
>
> *Q1*: My main concern is about the correctness of the derivation in the proposed method. The proof of Theorem 1 says both Equation (7), the objective and Equation (8), the cost constraint, are linear w.r.t. $\pi^{i_h}$, which looks strange to me. The equations involve advantage estimation which definitely has a non-linear relationship with  $\pi^{i_h}$. Why would it be linear?
>
> *A1*: We apologize for the confusion caused by the incorrect description in the proof of Theorem 1. As the policy for an agent is actually a distribution, it may be inappropriate to say the objective (Eq.7)and cost constraint (Eq.8) are linear w.r.t. $\pi^{i_h}$. We appreciate your attention to this matter. In fact, the core intention is to demonstrate that the Problem (7) and (8) are convex. Here, because $\boldsymbol{\pi_{\theta_k}}$ and $\theta_{k+1}^{i_{1:h-1}}$ are given, $E_{s \sim \rho_{\boldsymbol{\pi_{\theta_k}}},a^{i_{1:h-1}}\sim \pi_{\theta_{k+1}}^{i_{1:h-1}},a^{i_h}\sim \pi^{}i_h}[A_{\boldsymbol{\pi_{\theta_k}}}^{i_h}(s, a^{i_{1:h-1}}, a^{i_h})]$ is similar to $E_{s \sim \rho_{\boldsymbol{\pi_{\theta_k}}},a^{i_h}\sim \pi^{i_h}}[A_{j,\boldsymbol{\pi_{\theta_k}}}^{i_h}(s, a^{i_h})]$ so that we only consider the latter one. We can divide the formula like this: $E_{s \sim \rho_{\boldsymbol{\pi_{\theta_k}}},a^{i_h}\sim \pi^{i_h}}[A_{j,\boldsymbol{\pi_{\theta_k}}}^{i_h}(s, a^{i_h})] =\sum_s \rho_{\boldsymbol{\pi_{\theta_k}}}(s) \sum_{a^{i_h}} \pi^{i_h}(a^{i_h}|s) A_{j,\boldsymbol{\pi_{\theta_k}}}^{i_h}(s, a^{i_h})$, where $\rho_{\boldsymbol{\pi_{\theta_k}}}(s)$ represents state visitation frequencies. We easily know that $\rho_{\boldsymbol{\pi_{\theta_k}}}(s)$ is not affected by $\pi^{i_h}$. Similarly, for each action of agent $i_h$, the $A_{j,\boldsymbol{\pi_{\theta_k}}}^{i_h}(s, a^{i_h})$ is also not relative to $\pi^{i_h}$. To this end, when $\boldsymbol{\pi_{\theta_k}}$ and $\theta_{k+1}^{i_{1:h-1}}$ are given, $E_{s \sim \rho_{\boldsymbol{\pi_{\theta_k}}},a^{i_h}\sim \pi^{i_h}}[A_{j,\boldsymbol{\pi_{\theta_k}}}^{i_h}(s, a^{i_h})]$ is convex to $\pi^{i_h}$. The same analysis applies to Equation (7) as well, confirming the convexity of both the objective and the cost constraint.  What's more, the good performance can also show the correctness of our algorithm to some degree.
>
> *Q2*:  When solving the optimization problem, in Corollary 2, the expectation of the advantage over the optimal policy $\pi^{i_h*}$ is assumed to be zero. The advantage means the performance improvement of the optimal policy. How can it be reasonable to assume it to be zero?
>
> *A2*: We are sorry that we may have omitted some derivation process in the manuscript. As we employ an intuitive heuristic based on primal-dual gradient methods [1], we apply the gradient descent method to estimate $\nu_j$ during the optimization for agent $i_h$. As for the last term in the gradient expression, direct evaluation may be infeasible since $\pi^{i_h*}$ may not locate in parameterized policy space. However, we can know that the optimal policy $\pi^{i_h*}$ satisfies the constraint of Equation (3). This allows us to assume that $\pi^{i_h*}$ and $\pi_{\theta_k}^{i_h}$ are very close. Following the definitions in reinforcement learning, we can know $A_\pi(s,a)=Q_\pi(s,a)-V_\pi(s)$ and $V_\pi(s)=\sum_a \pi(s,a) Q_\pi(s,a) =E_{a \sim \pi}[Q_\pi(s,a) ]$, we can easily know the $E_{a \sim \pi}[A_\pi(s,a)]=\sum_a \pi(s,a) A_\pi(s,a)=\sum_a \pi(s,a) Q_\pi(s,a)-\sum_a \pi(s,a) V_\pi(s)=\sum_a \pi(s,a) Q_\pi(s,a) -V_\pi(s)=0$. To this end, we can adopt $\pi_{\theta_k}^{i_h}$ to approximate the expectation of advantage over $\pi^{i_h*}$ and get $E_{s \sim \rho_{\boldsymbol{\pi_{\theta_k}}},a^{i_h} \sim \pi^{i_h*}}[A_{j,\boldsymbol{\pi_{\theta_k}}}^{i_h}(s, a^{i_h})] \approx E_{s \sim \rho_{\boldsymbol{\pi_{\theta_k}}},a^{i_h} \sim \pi_{\theta_k}^{i_h}}[A_{j,\boldsymbol{\pi_{\theta_k}}}^{i_h}(s, a^{i_h})]=0$.
>
> This approximation may bring some error so that we also introduce a per-state acceptance indicator function $I(s_j)=\boldsymbol{1}\_{D\_{KL}(\pi\_\theta^{i_h},\pi\_{\theta_k}^{i_h}) \leq \delta}$ to enforce the constraint that updated policy remains close to $\pi_{\theta_k}^{i_h}$. Our experimental results also demonstrate that the proposed approximation is reasonable and leads to satisfactory performance in practice.
>
> Finally, thank you again for your comments. We will incorporate your suggestions and describe our method more clearly in our next revision. If some of your concerns are addressed, you could consider raising the rating. This is very important for us and we will appreciate it very much.
>
> [1] D. P. Bertsekas, Constrained optimization and Lagrange multiplier methods. Academic press, 2014.

---

> > ### Comment · Reviewer_eMiG · 2023-08-21
> > **Similar to the FOCOPS paper**
> >
> > Thanks for the rebuttal. However, after reading the response and the cited works in the paper more detailedly, I find the paper has mostly repeated the theory from the single-agent FOCOPS paper [1] while only slightly mentioning the paper. Roughly, the paper can be summarized as applying the first-order optimization method in [1] to the MARL framework in [2]. Moreover, the multiple importance ratio introduced in [2] may actually incur unstable and degraded performance.
> >
> > As such, I would not recommend acceptance of the paper to the NeurIPS venue.
> >
> >
> > [1] Zhang, Yiming, Quan Vuong, and Keith Ross. "First order constrained optimization in policy space." Advances in Neural Information Processing Systems 33 (2020): 15338-15349.
> >
> > [2] Kuba, Jakub Grudzien, et al. "Trust region policy optimisation in multi-agent reinforcement learning." arXiv preprint arXiv:2109.11251 (2021).

---

> > > ### Author Response · Authors · 2023-08-21
> > >
> > > Thanks for the comment. As for the multiple importance ratio, we think the reviewer may refer to the $M^{i_{1:h}}(s, \pmb{a})$ in [1]. In fact, it is derived by the sequential update scheme to model the correlations among agents in multi-agent problems, which has been adopted by quite a few works [1, 2, 3, 4]. This framework has been rigorously proven and shows good performance in solving multi-agent problems. In this way, we believe it can help deal with the challenges in MARL instead of incurring unstable and degraded performance.
> > > In addition, we would like to emphasize that our manuscript aims to extend and apply the first-order optimization method to the  safe multi-agent problems, where there are  very few studies. We believe our work offers new insights into the challenges and opportunities presented by multi-agent settings, which are considerably more complex than single-agent scenarios.
> > >
> > > [1] J. G. Kuba, R. Chen, M. Wen, Y. Wen, F. Sun, J. Wang, and Y. Yang, “Trust region policy optimisation in multi-agent reinforcement learning,” in International Conference on Learning Representations, 2022.
> > >
> > > [2] Wen, Muning, et al. "Multi-agent reinforcement learning is a sequence modeling problem." Advances in Neural Information Processing Systems 35 (2022): 16509-16521.
> > >
> > > [3] S. Gu, J. G. Kuba, Y. Chen, Y. Du, L. Yang, A. Knoll, and Y. Yang, “Safe multi-agent reinforcement learning for multi-robot control,” Artificial Intelligence, p. 103905, 2023.
> > >
> > > [4] Kuba, Jakub Grudzien, et al. "Heterogeneous-agent mirror learning: A continuum of solutions to cooperative marl." arXiv preprint arXiv:2208.01682 (2022).

---

### Official Review · Reviewer_fTdt · 2023-07-06

**Soundness:** 3 good
**Presentation:** 2 fair
**Contribution:** 3 good
**Rating:** 6
**Confidence:** 2

**Summary:**

The paper builds over previous work (1) to construct a safe policy optimization algorithm based on first-order methods. It shows empirical improvement over some baselines.

**Strengths:**


# quality

The quality of this work is fairly satisfactory, as all the needed theory is introduced and a practical counterpart is proposed.

# clarity

The exposition is clear





**Weaknesses:**

# originality
The work seems not to be much original since it consists of a fair extension of a previously cited paper.

# significance

I might say that the work does not seem significant enough compared to related works. For instance, one positive fact referred to in the abstract (upper bound violation) is a direct property of using the previous algorithm with an additional constraint.


**Questions:**

I don't have questions

**Limitations:**

I didn't find any section where the limitations of the proposed method were addressed.

---

> ### Author Rebuttal · Authors · 2023-08-09
>
> *Q1*: The work seems not to be much original since it consists of a fair extension of a previously cited paper.
>
> *A1*: We believe our work is not merely an extension of a previously cited paper, which we think you refer to MACPO. As is discussed in the paper, although MACPO offers an impressive solution for safe multi-agent reinforcement learning, it does owns some limitations. For example, achieving parameterized policies involves second order optimization, resulting in complex computation and implementations. Furthermore, the approximation during the optimization process will bring nonnegligible approximation error, necessitating additional steps during each update in the training process to recover from constraint violations. In contrast, our work proposes a fundamentally different way to tackle the optimization problem, where we provide full derivatives. We design a two-step approach to solve the optimization problem and prove that we can successfully employ first order method for the optimization. This significant departure from MACPO's methodology sets our work apart from the previously cited paper.
> To reinforce the distinctiveness of our contribution, we conduct extensive experiments to compare our algorithm with MACPO, from the performance to the computing efficiency. The results from these experiments also highlight the outstanding performance of our algorithm. To sum up, we think our work offers a novel and distinct contribution to the field of safe multi-agent reinforcement learning, as evidenced by the differences in methodology, proofs, and experimental results when compared to the previously cited paper.
>
> *Q2*: I might say that the work does not seem significant enough compared to related works. For instance, one positive fact referred to in the abstract (upper bound violation) is a direct property of using the previous algorithm with an additional constraint.
>
> *A2*: As is introduced in our work, safe multi-agent reinforcement learning holds vital significance as many applications often require the agents to refrain from taking certain actions or visiting particular states in real world. Among related research, MACPO achieves the state-of-the-art performance. But compared to this second-order method, our work not only achieves superior performance but also improve the computation efficiency, with rigorous theoretical guarantees and relying exclusively on first-order optimization techniques. These advantages are crucial, particularly when dealing with large state/action spaces and complex scenarios. Regarding the mentioned upper bound violation, it serves as an important security guarantee in safe reinforcement learning, ensuring the degree of constraint violation can be controlled within a certain range even in  worst-case scenarios. It should be noted that this property is not a direct consequence of using an additional constraint as the addition of such a constraint could also lead to potential violations. Taking MAPPO-Lagrangian as an example, as a soft constraint algorithm, whether this algorithm satisfies any worst-case constraint guarantees remains  unknown. The empirical results in Safe Multi-Agent-Isaac-Gym obviously shows that it exhibits a delay in guaranteeing safety and even a sudden drop in performance when enforcing the safety constraint. This indicates that the property of upper bound violation is important, which also demonstrating the significance of our work.
>
> Finally, thank you again for your comments. If some of your concerns are addressed, you could consider raising the rating. This is very important for us and we will appreciate it very much.

---

> > ### Comment · Reviewer_fTdt · 2023-08-18
> >
> > I would like to thank the Authors for the clarifications, I believe all my doubts have been clarified and I am raising my score. I would suggest adding a section (Maybe on the +1 page of the camera-ready version) that clarifies the relationship with MACPO as in the Authors' reply. It would be of high value, not only to help the reader understand the main point more clearly, but also because I believe some of the contributions stand in the step forward with respect to it.

---

> > > ### Author Response · Authors · 2023-08-18
> > >
> > > Thank you for your valuable feedback and for raising your score after considering our clarifications. We appreciate your suggestion and agree that adding clarifications about the relationship between our algorithm and MACPO would be of high value to the readers. We will incorporate your suggestion and include the suggested section in the revised version of the paper. Thank you once again for your thoughtful comments and for helping us improve our manuscript.

---

### Official Review · Reviewer_DYiL · 2023-07-11

**Soundness:** 3 good
**Presentation:** 3 good
**Contribution:** 3 good
**Rating:** 6
**Confidence:** 4

**Summary:**

The paper proposes a new method called first-order constraint optimization in multi-agent policy space (MAFOCOPS), which solves constraint optimization problems in a non-parametric policy space and then projects the updated policy back into the parametric policy space to achieve feasible strategies that meet safety constraints. Experimental results show that the proposed method has achieved remarkable performance while satisfying safe constraints on multiple safe MARL benchmarks.


**Strengths:**

The paper proposes a new method called Multi-Agent First Order Constrained Optimization in Policy Space (MAFOCOPS) to address the challenge of developing safety-aware methods for multi-agent reinforcement learning (MARL). The proposed approach solves the constrained optimization problem in a non-parametric policy space and then projects the updated policy back into a parametric policy space, ensuring feasible policies that satisfy safety constraints.

The MAFOCOPS method has first-order characteristics, making it relatively easy to implement compared to more complex optimization methods. Additionally, it provides an approximate upper bound for constraint violations in worst-case scenarios. Experimental results demonstrate that the MAFOCOPS method achieves significant performance improvements in multiple safety MARL benchmarks while simultaneously satisfying safety constraints. This indicates that the proposed approach effectively balances performance and safety considerations. Unlike existing safety MARL methods that are often tailored to specific tasks or rely on specific assumptions and techniques, the MAFOCOPS method is designed to be applicable to a wide range of multi-agent systems and tasks. This generalizability enhances the practicality and versatility of the proposed approach.


**Weaknesses:**

The performance of the method is sensitive to certain hyperparameters, such as the Lagrange multipliers and the safety bound. While the paper claims that the method is relatively insensitive to variations in these hyperparameter values, it would be helpful to provide a more detailed analysis of the sensitivity and its impact on performance.

The paper does not provide information on the implementation details of the proposed method, such as the specific algorithms used for policy optimization or the computational complexity of the method. Providing these details would help in assessing the practical feasibility and efficiency of the proposed approach.

**Questions:**

1. How can we select the best hyperparameter values for a new environment which hasn’t been met before, is there any search method like grid search or multi-arm bandit?

2. Is it possible to introduce more baseline algorithms for comparison?

3. I didn’t see a detailed explanation of whether MAFOCOPS uses Monte Carlo return or bootstrap return in the Safe-Multi-Agent-Isaac-Gym in the paper. I am curious if this factor will have a significant impact on the results. If possible, please provide experimental results using these different target returns.


**Limitations:**

The penultimate section of the paper contains a discussion of the main limitations. The method itself should not have a direct negative outcome. However, note that in practical applications they could occur. Some examples and ways to address them could be discussed.

---

> ### Author Rebuttal · Authors · 2023-08-09
>
> *Q1*:The performance of the method is sensitive to certain hyperparameters, such as the Lagrange multipliers and the safety bound. While the paper claims that the method is relatively insensitive to variations in these hyperparameter values, it would be helpful to provide a more detailed analysis of the sensitivity and its impact on performance.
>
> *A1*:Due to the page limit, the results of our sensitivity analysis is presented in the appendix. Considering the complexity of multi-agent environments, establishing a precise correlation between the performance of our algorithm the Lagrange multipliers $\lambda_j$ and $\nu_{max}$ is difficult. It can be observed that different scenarios have different sensitivity to these hyperparameters from the results. Overall, the reward achieved under different settings is relatively insensitive as even setting these parameters across a broad range only leads to an average degradation of less than 10\%. On the other hand, the cost may be more sensitive to these parameters,  highlighting the inherent challenges in ensuring safety guarantees in safe multi-agent reinforcement learning to some degree.
>
> As for the safety bound, we can see that setting it too high could lead to increased oscillations in cost performance, although it may yield better reward performance. This observation is reasonable since higher safety bounds may allow agents to explore actions with potentially higher returns but less safety. If the safety bound is low enough, the agents will take actions that are definitely safe, leading to less vibration. Whereas, from a global perspective, the effectiveness of our algorithm remains consistent across these different safety levels. In general, we need to strike a balance between ensuring safety and achieving a good reward performance when applying the method.
>
> *Q2*:The paper does not provide information on the implementation details of the proposed method, such as the specific algorithms used for policy optimization or the computational complexity of the method. Providing these details would help in assessing the practical feasibility and efficiency of the proposed approach.
>
> *A2*:Thank you for your advice and we will provide more analysis in future revision. In fact, we introduce some details about our algorithms in Section 4.3 and specific algorithms in the appendix. Here we also introduce some basic information for convenience. For the training process, after initialising the needed parameters and networks, we can utilize the models to generate data and then estimate the advantages. Then the update of $\nu_j$ is performed using Equation (10) in the manuscript. After that, we can update the value networks using Mean Square Error loss and update policy networks using the equations of $\nabla_\theta L(\theta)$, which is expressed in Equation (11). During the training, our algorithm also employs the early stopping criterion to ensure that the updated policy satisfies the trust region constraint. In addition, the computational complexity of our algorithm is $\mathcal{O}$$(NKMHP)$, where $N$ denotes the number of agents, $K$ denotes the number of update times in every epoch, $M$ denotes the number of training steps, $H$ denotes the number of constraints and $P$ denotes the number of parameters.
>
> *Q3*: How can we select the best hyperparameter values for a new environment which hasn't been met before, is there any search method like grid search or multi-arm bandit?
>
> *A3*: In our work, when we conduct the experiments in a new environment, we use grid search to select the hyperparameters, which is a commonly used method for hyperparameter tuning.
>
> *Q4*: Is it possible to introduce more baseline algorithms for comparison?
>
> *A4*: As we mentioned in our work, the realm of safe multi-agent reinforcement learning is a nascent area of research and there are limited related works available. What's more, most current algorithms have certain limitations or are tailed for robotics tasks, whose settings are very different from our work. In this way, we choose to adopt the state-of-the-art (SOTA) algorithms in this field, namely MACPO and MAPPO-Lagrangian [1], as the baseline algorithms, which are published in the well-known journal, Artificial Intelligence. We think that the comparison with these two baseline methods are convincing to prove the effectiveness of our algorithm.
>
> *Q5*: I didn't see a detailed explanation of whether MAFOCOPS uses Monte Carlo return or bootstrap return in the Safe-Multi-Agent-Isaac-Gym in the paper. I am curious if this factor will have a significant impact on the results. If possible, please provide experimental results using these different target returns.
>
> *A5*: Thank you for your advice. In fact, as is mentioned in our work, we use Generalized Advantage Estimator (GAE) [2] to estimate the advantages. As Monte Carlo return is known to have low bias but high variance while bootstrap return has low variance but high bias, GAE is a method that can achieve a balance, making it be widely adopted to estimate return in policy-based methods. Considering that the baseline algorithms both use this method for advantages estimation, we also utilize it based on its effectiveness and compatibility with our approach. As the hyperparameters of GAE remain the same across our experiments, we believe that the superior performance of our approach is sufficient evidence of its effectiveness.
>
> [1]  S. Gu, J. G. Kuba, Y. Chen, Y. Du, L. Yang, A. Knoll, and Y. Yang, “Safe multi-agent reinforcement learning for multi-robot control,” Artificial Intelligence, p. 103905, 2023.
> [2]  J. Schulman, P. Moritz, S. Levine, M. Jordan, and P. Abbeel, “High-dimensional continuous control using generalized advantage estimation,” arXiv preprint arXiv:1506.02438, 2015.

---

> > ### Comment · Reviewer_DYiL · 2023-08-21
> >
> > Thanks to the authors for providing detailed answers to the questions I raised, but due to the fact that the performance demonstrated by multi-agent reinforcement learning (MARL) algorithms mostly relies on the tuning of parameters, the addition of safety settings to the MARL algorithm has further improved the level of "tricky" tuning. I actually prefer to see the author present more ablation experimental results under different hyperparameter search strategies. Based on the above point of view, I choose to keep my score.

---

> > > ### Author Response · Authors · 2023-08-21
> > >
> > > Thank you for your feedback and comments. We understand your interest in seeing more ablation experimental results under different hyperparameter search strategies in the research of MARL. In fact, there are few works that pay attention to the tuning of parameters in MARL while many works still show good effectiveness. We agree that this is a valuable question and we would conduct more experiments and research about this field in future.

---

### Author Rebuttal · Authors · 2023-08-09

Thank all reviewers for your time and valuable suggestions. We hope our rebuttal could address your concerns. We would appreciate it if you could re-evaluate our submission and we are looking forward to discussions if you have any other concerns.

---

### Decision · Program_Chairs · 2023-09-21

**Decision:**

Accept (poster)

**Comment:**

The paper proposes Multi-Agent First Order Constrained Optimization in Policy Space (MAFOCOPS), an safety-aware algorithm for multi-agent reinforcement learning (MARL). The proposed approach solves the non-parametric constrained optimization problem and projects the policy back into a parametric policy space to enforce safety. The idea of this algorithm is quite interesting and novel in the MARL domain. The first-order method implemented is also practical and easy enough to implement, with experimental results showing significant performance improvements. I agree with most reviewers that the studied problem is important and the paper is by far quite well-written.

Some reviewers express concerns on the theoretical contribution, which makes this paper ending up with an overall borderline score. But on the overall, I still think the merits of this paper (problem motivation/formulation, proposed algorithms, experiments) outweigh the downsides raised and therefore recommend acceptance